# TerraBind: Fast and Accurate Binding Affinity Prediction through Coarse Structural Representations

Matteo Rossi [* 1]  Ryan Pederson [* 1]  Miles Wang-Henderson [1]  Ben Kaufman [1]  Edward C. Williams [1]
Carl Underkoffler [1]  Owen Lewis Howell [1]  Adrian Layer [1]  Stephan Thaler [1]  Narbe Mardirossian [1]
John Anthony Parkhill [1]

## Abstract

We present TerraBind, a foundation model for protein-ligand structure and binding affinity prediction that achieves $26\times$ faster inference than state-of-the-art methods while improving affinity prediction accuracy by up to 20%. Current deep learning approaches to structure-based drug design rely on expensive all-atom diffusion to generate 3D coordinates, creating inference bottlenecks that render large-scale compound screening computationally intractable. We challenge this paradigm with the hypothesis: full all-atom resolution is unnecessary for accurate small molecule pose and binding affinity prediction. TerraBind tests this hypothesis through a coarse pocket-level representation (protein $C_\beta$ atoms and ligand heavy atoms only) within a multimodal architecture combining pretrained molecular encoders and ESM-2 protein embeddings that learns rich structural representations, which are used in a diffusion-free optimization module for pose generation and a binding affinity likelihood prediction module. On structure prediction benchmarks, TerraBind matches diffusion-based baselines in ligand pose accuracy. For binding affinity, TerraBind outperforms Boltz-2 by 16–20% in Pearson correlation on both a public benchmark (CASP16) and a diverse private dataset (18 assays). The affinity module also provides well-calibrated uncertainty estimates, addressing a critical gap in compound prioritization for drug discovery. Furthermore, this module enables a continual learning framework and a hedged batch selection strategy that, in simulated drug discovery cycles, achieves $6\times$ greater affinity improvement over greedy approaches.

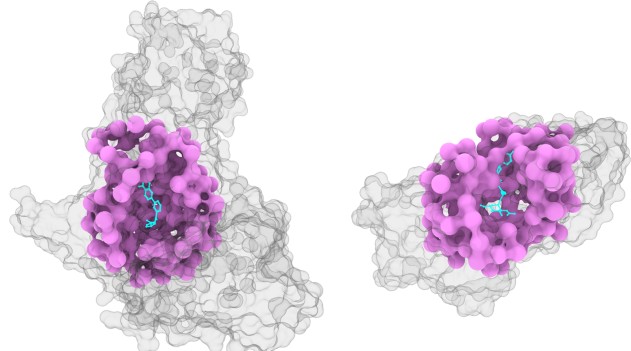

*Figure 1.* **Structure predictions for PDB 8JJT and 8B1C.** Purple regions show predicted binding site residues (within 15 Å of any ligand atom) overlaid on the gray ground truth structure. Proteins are represented by $C_\beta$ atoms; ligands by all heavy atoms.

[*]Equal contribution  [1]Terray Therapeutics, Inc., Monrovia, CA, USA. Correspondence to: Matteo Rossi <matteo.rossi@terraytx.com>.

*Proceedings of the $43^{rd}$ International Conference on Machine Learning*, Seoul, South Korea. PMLR 306, 2026. Copyright 2026 by the author(s).

## 1. Introduction

Accelerating the discovery of small molecule therapeutics requires computational models that balance high predictive accuracy with the throughput necessary to screen vast chemical spaces. Traditionally, structure-based drug design (SBDD) has relied on physics-based docking (Wang et al., 2016) (e.g., Glide (Friesner et al., 2004), AutoDock Vina (Eberhardt et al., 2021)), which is computationally efficient but often fails to capture induced-fit effects and suffers from limited scoring accuracy. The landscape has been fundamentally shifted by deep learning models—such as AlphaFold2 (Jumper et al., 2021), RoseTTAFold (Baek et al., 2021), AlphaFold3 (Abramson et al., 2024), NeuralPLexer3 (Qiao et al., 2025), and Boltz (Wohlwend et al., 2024; Passaro et al., 2025)—which jointly fold (co-fold) proteins and ligands. While these models achieve structural fidelity and affinity predictions approaching the accuracy of physics-based free energy calculations, this performance comes at a prohibitive computational cost. State-of-the-art systems like Boltz-2 rely on iterative diffusion processes that require approximately 20 seconds per complex (Passaro et al., 2025), rendering them impractical for high-throughput binding affinity prediction.

Recent attempts to mitigate this latency often compromise accuracy or generalizability. Hybrid approaches like Boltzina (Furui & Ohue, 2025) reintroduce rigid-pocket assumptions, while sequence-based models (Wang-Henderson et al., 2025; Khokhlov et al., 2025) offer speed but struggle on data-poor targets. This creates a fundamental tension: scalable methods often lack structural nuance and generalizability, while accurate diffusion-based models are computationally intractable for industrial discovery.

Furthermore, deployment is hindered by a lack of rigorous uncertainty quantification for binding affinity. Structural confidence scores (e.g., ipTM and pLDDT) assess geometric plausibility rather than affinity reliability (Passaro et al., 2025); critically, a model may generate a geometrically confident pose yet remain entirely uncertain about the magnitude of the binding interaction. Existing uncertainty models often lack target generality or integration into co-folding workflows (Luo et al., 2023), and joint uncertainty quantification for batches—shown to facilitate robust molecule selection in DMTA cycles (Wang-Henderson et al., 2025)—remains absent from current architectures.

To simultaneously resolve these inference bottlenecks and reliability gaps, we challenge the necessity of the computationally intensive components inherent to current co-folding architectures. We posit that the expensive diffusion-based generation used in state-of-the-art models is not only a bottleneck for throughput but is also largely superfluous for the core task of binding affinity prediction. This work is motivated by a critical hypothesis: *full all-atom diffusion is not required to accurately predict binding affinity, nor is it strictly necessary for recovering high-fidelity ligand poses.* We propose that rich coarse structural representations can capture the essential geometric and chemical information needed for both structure and affinity prediction without the overhead of explicit generative modeling (Fig. 1).

To this end, we introduce **TerraBind** (Fig. 2), a foundation model for protein-ligand structure and binding affinity prediction designed for high-throughput applications—from virtual screening to hit optimization. Our approach focuses on learning rich structural representations at the binding site interface to directly inform affinity prediction, while employing a lean architecture—only $\sim$30M trainable parameters compared to $\sim$509M for Boltz-2—to minimize computational cost without sacrificing accuracy. Our key contributions are:

- A diffusion-free structure prediction module that combines pretrained encoders COATI-3 for ligands, ESM-2 for proteins (Lin et al., 2023)) with a lean 48-layer Pairformer architecture. It achieves $26\times$ faster inference than Boltz-2 while matching pose accuracy on Fold-Bench, PoseBusters, and Runs N' Poses.
- An intrinsic structural uncertainty via pairwise distance

entropy $H_{\text{LP}}$ (Eq. 14), which correlates with pose accuracy without requiring an auxiliary confidence module.
- An affinity module outperforming Boltz-2 on CASP16 (+16% Pearson) and 15/18 proprietary targets (+20%).
- A structural fine-tuning on crystal data (3–6 examples) yields a $\sim$17% affinity gain on held-out compounds.
- An epistemic neural network (epinet) module for calibrated affinity uncertainty, where lower uncertainty scores correlate with higher prediction success.
- Continual learning for DMTA that leverages epinet uncertainty to enable hedged batch selection, outperforming greedy strategies by $6\times$ in affinity improvement.

**Conflict of Interest Disclosure.** This work was conducted at Terray Therapeutics, which leads the development of TerraBind, the model presented and evaluated in this paper. All authors were employed by Terray Therapeutics during the period this work was conducted. S.T. joined Terray Therapeutics in January 2026 and previously contributed to the development of Boltz-2, one of the baseline models against which TerraBind is compared in this work. The proprietary benchmark assays used in this paper (Section 3.2) were generated internally at Terray Therapeutics.

## 2. Methods

The TerraBind architecture (Fig. 2) consists of four primary components: (1) Frozen pretrained encoders (a multimodal molecular encoder and ESM-2) that provide initial chemical and biological embeddings; (2) a structure module with a 48-layer Pairformer trunk that learns pocket-level representations via pairwise distance prediction; (3) a pose module for coordinate generation; and (4) an affinity module that maps structural features to binding strength, utilizing an epistemic neural network for calibrated uncertainty.

### 2.1. Frozen Pretrained Encoders

**Multimodal Molecular Encoder:** We pretrain COATI-3, our multimodal molecular encoder, which extends COATI (Kaufman et al., 2024a;b) and integrates three chemical modalities: (i) a text transformer for SMILES strings, (ii) a graph transformer for 2D molecular graphs, and (iii) a chirality-aware $SE(3)$-equivariant Allegro-based neural network (Geiger & Smidt, 2022; Musaelian et al., 2023) for 3D conformer point clouds. The model was pretrained using a contrastive loss on over a billion compounds from diverse sources, including USPTO (Lowe, 2012), BindingDB (Gilson et al., 2016), Buchwald-Hartwig (Ahneman et al., 2018), SAVI (Patel et al., 2020), COCONUT (Sorokina et al., 2021), PCCL (Bedart et al., 2024), and Plinder (Durairaj et al., 2024). By aligning these modalities, the encoder provides high-resolution, atom-level embeddings that capture a chemical space far more vast than the $\sim$50,000 unique

ligands found in experimental structures.

**Protein Encoder:** We use ESM-2 (650M) (Lin et al., 2023), a masked language model trained on ∼65M sequences. ESM-2 extracts evolutionary information and implicit structural propensities directly from sequences without requiring multiple sequence alignments (MSAs). Utilizing these frozen pretrained residue embeddings provides a more robust global context and significantly higher sample efficiency than learning protein representations from scratch on limited structural data.

## 2.2. Structure Module

The structure module predicts 3D geometric relationships between atoms in protein-ligand complexes. It processes frozen encoder outputs through a trainable Pairformer trunk to predict categorical distributions over pairwise distance bins, trained with cross-entropy loss (Fig. 2).

### 2.2.1. PAIRFORMER TRUNK

The architecture is adapted from (Abramson et al., 2024), utilizing triangle attention and multiplication (see Appendix A.1) to update pair representations. Key modifications for efficiency include: (1) removing the single-sequence representation and downstream diffusion module, and (2) utilizing frozen molecular ($SE(3)$-equivariant) and protein embeddings instead of trainable encoders. These changes reduce the parameter count from 147M to 27M for a 48-layer trunk. The model is trained via categorical cross-entropy over 64 distance bins ($< 2$Å to $> 22$Å) for all ligand heavy atoms and protein residue centers ($C_\beta$), with pair-type weighting (see Appendix A.2.2 for the full loss formulation). From these distributions, we derive the expected distance and normalized ligand-protein pairwise entropy ($H_{\text{LP}}$) (see Appendix A.2.3, A.2.4). The expected distances define a 15Å binding pocket (Eq. 11) that serves as input for the pose and affinity modules. $H_{\text{LP}}$ serves as both a structural uncertainty estimate and a zero-shot binding affinity signal.

### 2.2.2. TRAINING DATA AND PROTOCOL

Training data is filtered to retain only protein and small-molecule complexes, excluding nucleic acids (DNA/RNA), single ion ligands, or other non-drug-like entities. We utilize two primary sources: (1) **Experimental structures** from the Protein Data Bank (PDB) (wwPDB consortium, 2019) released before 2021-09-30 (167,588 complexes); and (2) **Distillation data** to increase coverage, including 542,378 high-confidence AlphaFold2 Swiss-Prot monomers (pTM > 0.9) (Jumper et al., 2021; Varadi et al., 2024) and 438,957 computationally generated BindingDB complexes via Boltz-1x (pIC50 > 6, ipTM > 0.9) (Wohlwend et al., 2024).

We employ a 105k-step three-stage curriculum with a batch size of 128 using mixed-precision training and cuEquivariance kernels (NVIDIA, 2024) on 4 H100 GPU nodes. The model first pretrains for 70k steps on diverse interface types using 384-token crops to establish structural priors. We then transition to ligand-centered 256-token crops for 20k steps using PDB and BindingDB data, and finally fine-tune the model for 15k steps exclusively on experimental PDB structures to prioritize ground-truth fidelity (see Appendix Table 1 for details). This focus on 256-token crops reflects drug discovery realities; binding pocket contexts—ligand atoms and residues within 15Å—rarely exceed 200 tokens (see Appendix Fig. 14). As TerraBind lacks a diffusion module to compensate for representational deficiencies, the Pairformer must capture all binding-relevant geometry directly. Combined with compact crops and the absence of a diffusion and confidence module, these choices yield a ∼2× reduction in training time relative to all-atom methods.

### 2.2.3. STRUCTURAL BENCHMARKS

We adopt the standard September 2021 training cutoff to facilitate comparison across existing approaches and compare TerraBind against Boltz-1 (Wohlwend et al., 2024) across four configurations: (1) TerraBind, utilizing full protein context; (2) TerraBind Pocket, which uses a two-stage inference (full protein forward pass to identify the binding pocket, followed by refined prediction on a 196-token local context, see Appendix A.4) to achieve $O(N^3)$ computational savings for high-throughput screening; (3) Boltz-1 Trunk, applying our coordinate optimization to Boltz-1 representations to isolate trunk quality; and (4) Boltz-1, the full diffusion-based pipeline. For each, we generate 10 samples, selecting the best pose via optimization loss (TerraBind/Trunk) or ipTM (Boltz-1). Performance is measured by ligand RMSD $< 2$Å success rate and a strict joint metric (RMSD $< 2$Å & LDDT-PLI $> 0.8$).

Generalization is assessed across four datasets: FoldBench ($n = 556$), a low-homology test set (Xu et al., 2026); PoseBusters ($n = 307$), curated for drug-like quality (Buttenschoen et al., 2024); Runs N' Poses ($n = 2,687$), a high-resolution zero-shot co-folding benchmark (Škrinjar et al., 2025); and a private validation set of co-crystal structures from active drug discovery programs. All benchmarks are filtered to exclude nucleic acids and non-drug-like entities.

## 2.3. Coarse Pose Module

To evaluate pose quality via standard metrics like RMSD, we employ a parameter-free optimization routine that generates 3D point clouds from the Pairformer's coarse-grained distograms (protein $C_\beta$ centers and ligand heavy atoms). We initialize coordinates from random noise and minimize the discrepancy between coordinate-derived distances and the

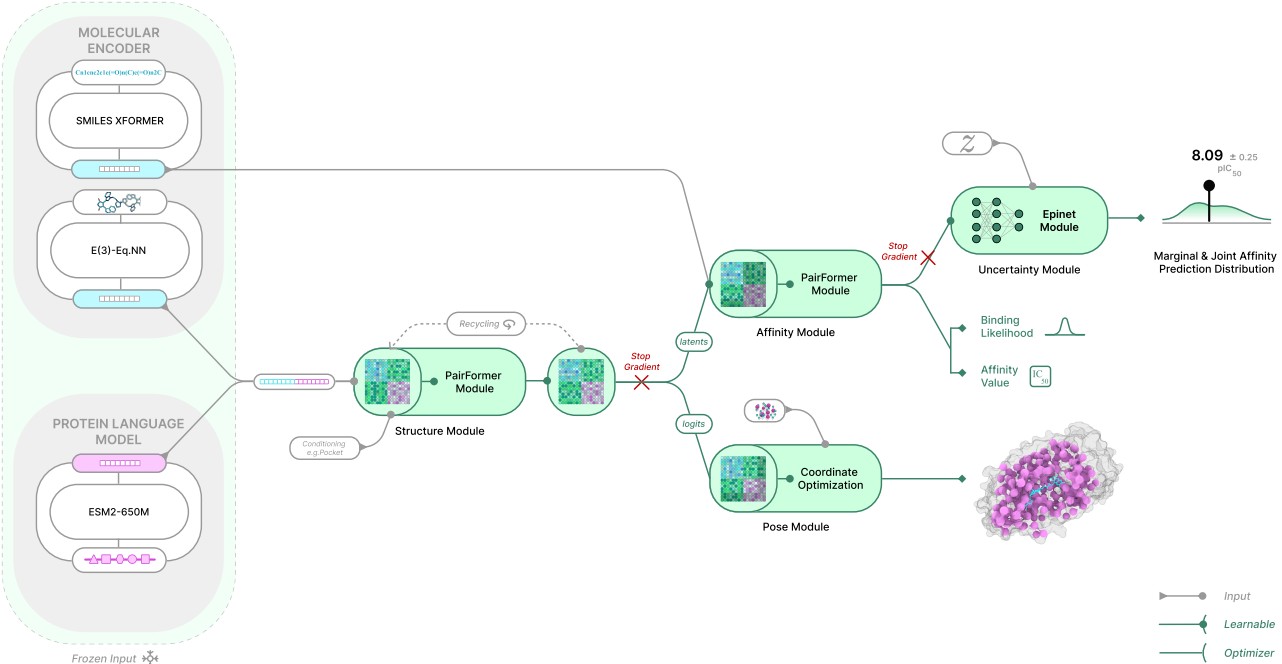

*Figure 2.* **TerraBind Architecture.** (1) **Frozen Encoders**: Multimodal molecular and ESM-2 protein encoders. (2) **Structure Module**: A 48-layer Pairformer predicts distance bin distributions, with pairwise entropy quantifying structural uncertainty. (3) **Pose Module**: Gradient-based optimization generates 3D coordinates from distance logits without diffusion. (4) **Affinity Module**: A 6-layer Pairformer utilizes structural features for binary and continuous affinity prediction, using an epistemic neural network for calibrated uncertainty.

predicted expected distances (see Appendix A.5).This approach follows an implicit energy-based paradigm (Wang & Du, 2025; Roney et al., 2025), converging in a few hundred steps (see Figure 1 for an example pose). The optimization is restricted to ligand heavy atoms and protein residue centers within 15Å of the ligand, matching the pocket definition used throughout the pipeline. This routine is highly parallelizable and more efficient than diffusion-based generation. Crucially, these 3D coordinates are used solely for visualization and structural evaluation; in TerraBind, affinity prediction operates directly on the latents and distance distributions, bypassing coordinate generation entirely.

### 2.4. Binding Affinity Module

TerraBind utilizes a specialized affinity module to map structure to binding potency. To enable modularity, the module is trained with frozen structural Pairformer latents.

#### 2.4.1. AFFINITY ARCHITECTURE

The affinity module is a 6-layer Pairformer that maps structural and chemical features to binding potency. Unlike Boltz-2, which conditions on distograms derived from diffused 3D coordinates (Passaro et al., 2025), we condition directly on the pairwise distance bin probabilities from the structural Pairformer to preserve geometric uncertainty. The input representation integrates four key components: (1)

structural Pairformer latents, (2) pairwise distance bin probabilities, (3) per-residue ESM-2 embeddings, and (4) an aggregated ligand embedding from the COATI-3 SMILES transformer.

To optimize efficiency, we restrict the input context to protein residues within 15Å of the ligand based on the expected distance pocket definition (see Appendix A.2.3) and mask protein-protein pairs. The resulting pair latents are mean-pooled into a complex representation $\mathbf{g}$, which feeds two MLP heads to predict binding likelihood $p_{\text{bind}}$ and quantitative affinity $\hat{y}$ in $\log_{10}$ units:

$$p_{\text{bind}} = \sigma(f_{\text{cls}}(\mathbf{g})) \in [0, 1], \qquad \hat{y} = f_{\text{reg}}(\mathbf{g}) \in \mathbb{R} \quad (1)$$

#### 2.4.2. AFFINITY LIKELIHOOD

To provide quantitative posterior estimates, we extend the affinity architecture with an epistemic neural network (epinet) (Osband et al., 2023; Wang-Henderson et al., 2025). The epinet is a lightweight MLP that takes the frozen complex latent $\mathbf{g}$ and an epistemic index $\mathbf{z} \sim \mathcal{N}(\mathbf{0}, \mathbf{I})$ sampled from a unit Gaussian to predict a residual $r_\theta(\mathbf{g}, \mathbf{z})$, yielding the sampled affinity:

$$\hat{y} \sim \hat{y}_{\text{TB}} + r_\theta(\mathbf{g}, \mathbf{z}) \quad (2)$$

where $\hat{y}_{\text{TB}}$ is the scalar quantitative prediction from the base affinity module and $r_\theta(\mathbf{g}, \mathbf{z}) = f_\theta(\mathbf{g}, \mathbf{z}) + f_\phi(\mathbf{g}, \mathbf{z})$ is

the learned residual consisting of a trainable component $f_\theta$ and a frozen prior network $f_\phi$ (Osband et al., 2023; Wang-Henderson et al., 2025). At inference time, we apply Eq. 2 by sampling multiple indices $\{\mathbf{z}_i\}_{i=1}^K$ (with $K = 1000$) to generate samples from the binding affinity posterior $p(y \mid \mathbf{g})$. Unlike large ensembles, this sampling carries negligible overhead as the expensive structural and affinity Pairformer latents do not need to be recomputed for each index. These posterior statistics quantify prediction uncertainty: complexes near the training manifold exhibit lower sample variance, reflecting higher certainty. Furthermore, the joint posterior between $B$ inputs allows us to model the correlation structure required for risk-aware molecular selection. We leverage the Expected Maximum (EMAX) acquisition function to maximize affinity within a batch:

$$\alpha_{\text{EMAX}} = \mathbb{E}_{\hat{y}_{1:B} \sim p(y_{1:B})}[\max(\hat{y}_{1:B})] \quad (3)$$

This framework also supports a *continual learning* scheme to rapidly incorporate new experimental observations without full model retraining (see Appendix A.6).

### 2.4.3. AFFINITY TRAINING DATA AND PROTOCOL

Our affinity training set prioritizes curated data from BindingDB (Liu et al., 2007), MF-PCBA (Buterez et al., 2023), PubChem (Kim et al., 2023), ChEMBL (Zdrazil et al., 2024), and the CeMM fragment screening dataset (Offensperger et al., 2024), supplemented by $\sim$1.2M synthetic decoys. Synthetic decoys are generated by sampling high-potency compounds ($< 1\mu$M) from ChEMBL and BindingDB, assuming they are target-selective, and assigning them as nonbinders for unrelated target sequences in the training data pool (Passaro et al., 2025). To ensure data quality, we apply a structure-uncertainty prefiltering strategy using $H_{\text{LP}}$ (see Appendix A.2.4). We exclude entries that exhibit high experimental potency ($< 1\mu$M) but lack a clear structural signal ($H_{\text{LP}} > 0.7$), as these often represent noise in structure-based learning. All experimental values ($K_i, K_d$, $\text{IC}_{50}$, etc.) are unified into a $\log_{10}$ scale. During training, we utilize precomputed structure embeddings and jointly supervise binary classification and quantitative affinity using a focal loss (Lin et al., 2017) and Huber loss (Huber, 1964), respectively. The Huber loss is applied to both absolute affinity values and pairwise intra-assay differences; the latter relative loss is critical for normalizing across disparate assay conditions (Passaro et al., 2025). For the synthetic decoys, we supervise both heads; the negative compounds are assigned a quantitative target of $> 100\,\mu$M ($\text{pIC}_{50} < 4$), and the quantitative loss is applied only when the model predicts a stronger potency than this threshold ($\text{pIC}_{50} > 4$). To manage data diversity, we employ a custom batch sampler with uniform probability across assays. For quantitative assays, we sample five random complexes per assay. For binary High-Throughput Screening (HTS) data, we sample a 1:4

ratio of positive to negative complexes per assay to address extreme class imbalance. This procedure allows the model to learn both hit discovery (binary) and lead optimization (relative/absolute regression) signals.

### 2.4.4. AFFINITY LIKELIHOOD TRAINING PROTOCOL

The epinet is trained exclusively on quantitative affinity data, utilizing the same batch sampling strategy as the base affinity module but omitting the relative loss. For each step, a single epistemic index $\mathbf{z} \sim \mathcal{N}(\mathbf{0}, \mathbf{I}_{256})$ is sampled and applied across the batch. The training objective is a Huber loss ($\delta = 0.5$) on the residual:

$$\mathcal{L}_{\text{epinet}} = \text{Huber}\left(y,\ \hat{y}_{\text{TB}} + r_\theta(\text{sg}(\mathbf{g}), \mathbf{z});\ \delta = 0.5\right) \quad (4)$$

where $\text{sg}(\cdot)$ denotes a stop-gradient on the latent $\mathbf{g}$, ensuring the epinet does not perturb the pretrained representations.

### 2.4.5. AFFINITY BENCHMARKS

We evaluate binding affinity prediction across two categories to test model generalization. First, we utilize CASP16, the recent protein-ligand binding affinity challenge (targets L1000 and L3000) (Gilson et al., 2026). Second, we assess the performance on 27,075 unseen proprietary compounds across 18 diverse assays from select drug discovery programs at Terray Therapeutics. This dataset is orders of magnitude larger than existing public benchmarks and each assay contains $\geq$95 $\text{IC}_{50}$ data points, providing a rigorous test of prospective performance on varied chemical series in realistic drug discovery campaigns. In Appendix B.4 we quantify activity cliff metrics on these datasets and find that our proprietary dataset is more challenging than public alternatives.

## 3. Results

### 3.1. Structure Prediction

#### 3.1.1. PER-BENCHMARK PERFORMANCE

We evaluate TerraBind's structure prediction capabilities across three public benchmarks—FoldBench, PoseBusters, and Runs N' Poses—alongside proprietary crystal structures. For each complex, we generate 10 samples; we select the best pose for optimization-based methods (TerraBind, TerraBind Pocket, Boltz-1 Trunk) by the lowest optimization loss, while diffusion-based methods (Boltz-1) are selected by the highest ipTM confidence. Metrics are computed using residue centers ($C_\beta$) and ligand heavy atoms to match our coarse representation. We report two success rates: ligand RMSD $< 2$Å within a 15Å pocket context, and a stricter combined metric requiring both RMSD $< 2$Å and LDDT-PLI $> 0.8$. The latter, computed using a 6Å radius, captures high-resolution local accuracy at the immediate binding interface. As shown in Figure 3, TerraBind consis-

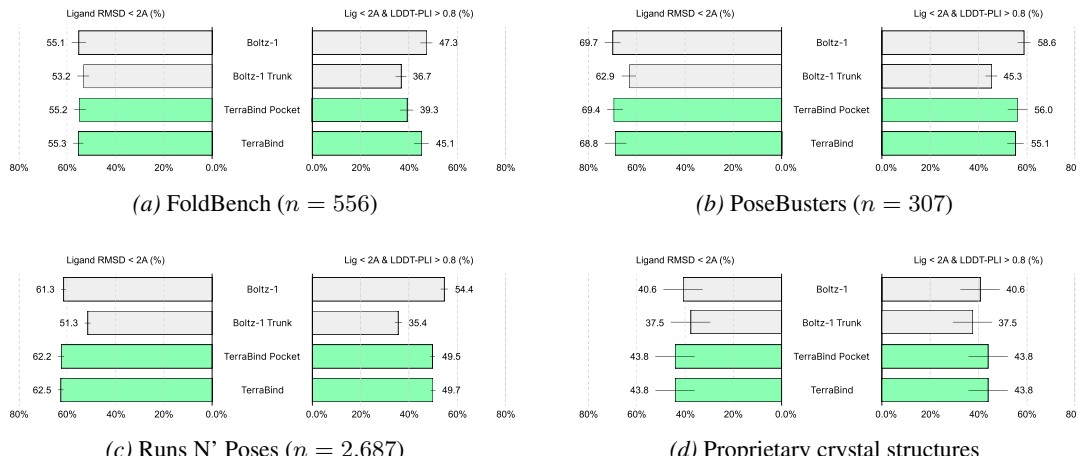

*(a)* FoldBench ($n = 556$)  *(b)* PoseBusters ($n = 307$)

*(c)* Runs N' Poses ($n = 2,687$)  *(d)* Proprietary crystal structures

*Figure 3.* **Structure prediction performance.** Each panel shows ligand RMSD $< 2$Å success rate (left bars) and combined success rate requiring both RMSD $< 2$Å and LDDT-PLI $> 0.8$ (right bars). (a) FoldBench, a low-homology benchmark; (b) PoseBusters, drug-like complexes; (c) Runs N' Poses, high-resolution systems; (d) Proprietary crystal structures from active drug discovery programs.

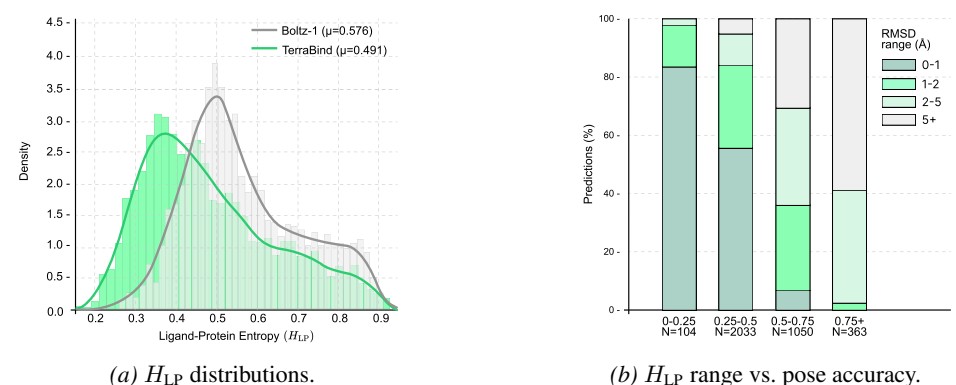

*(a)* $H_{LP}$ distributions.  *(b)* $H_{LP}$ range vs. pose accuracy.

*Figure 4.* **Distogram entropy as structural confidence.** (a) Aggregated distribution of $H_{LP}$ across benchmarks; TerraBind achieves 15% lower mean entropy than Boltz-1 Trunk, indicating more confident predictions. (b) Correlation between TerraBind entropy and pose accuracy, where lower entropy consistently predicts higher success rates across all RMSD thresholds.

tently achieves competitive accuracy compared to Boltz-1 without requiring a diffusion module. Notably, TerraBind significantly outperforms the Boltz-1 Trunk when using identical coordinate optimization, indicating that our Pair-former learns richer geometric representations. This improved structural fidelity at the 6Å interface likely underlies our superior binding affinity predictions (Section 3.2). Furthermore, the robust performance of TerraBind Pocket on a compact 196-token context confirms that binding-relevant information is concentrated within the pocket, validating our efficient ligand-centered training approach.

### 3.1.2. ENTROPY AS STRUCTURAL CONFIDENCE

Beyond pose accuracy, we evaluate whether the Pairformer's predicted distance distributions provide meaningful uncertainty estimates through the ligand-protein entropy ($H_{LP}$). As shown in Figure 4, TerraBind's entropy distribution is shifted significantly toward lower values compared to Boltz-

1 Trunk, achieving a mean $H_{LP}$ of $0.491$ versus $0.576$—a 15% reduction that indicates substantially more confident interface predictions. Crucially, $H_{LP}$ correlates strongly with pose accuracy: low-entropy predictions ($H_{LP} < 0.25$) consistently yield high success rates across accurate poses (RMSD $< 2$ Å), whereas high-entropy predictions ($H_{LP} > 0.75$) rarely do so. This monotonic relationship between entropy and accuracy emerges naturally from structure learning without explicit supervision, validating $H_{LP}$ as a reliable, model-intrinsic confidence metric that eliminates the need for a separate confidence module.

### 3.1.3. END-TO-END INFERENCE THROUGHPUT

TerraBind achieves a 26.6x end-to-end speedup over Boltz-2, generating 10 pose samples and the corresponding affinity prediction in 1.045 seconds per complex, compared to 27.8 seconds for Boltz-2 on equivalent hardware (single A6000 GPU, 196 tokens). This efficiency stems from our

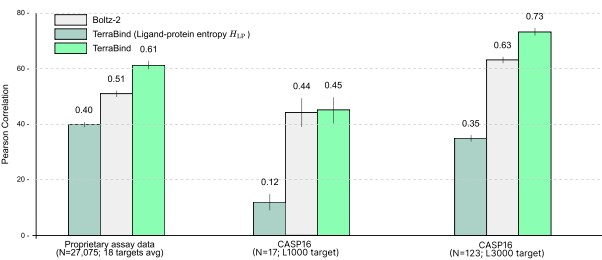

*(a)* Pearson correlation across benchmark datasets.

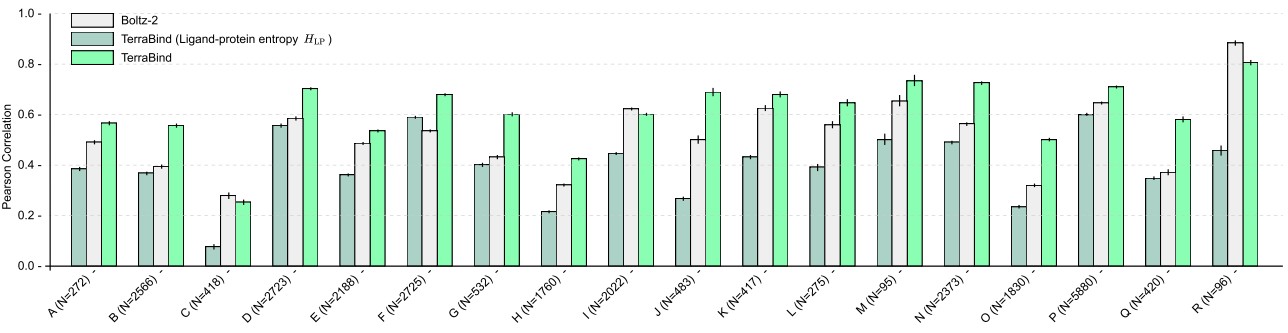

*(b)* Per-target correlation comparison across Proprietary assay data.

*Figure 5.* **Binding affinity prediction performance.** (a) Pearson correlation across benchmarks; Zero-shot ($H_{LP}$) also correlates with affinity. (b) Per-assay correlation showing TerraBind outperforms Boltz-2 on 15 of 18 diverse proprietary targets.

"diffusion-free" architecture; by bypassing the iterative generative coordinate-sampling bottleneck, structural latents are processed by the pose, affinity and uncertainty modules with negligible overhead. This enables simultaneous, high-throughput screening of 3D poses and binding potencies at a scale previously considered computationally intractable. A detailed decomposition of module-specific latencies is provided in Appendix B.1.

## 3.2. Binding Affinity Prediction

### 3.2.1. PER-BENCHMARK PERFORMANCE

We evaluate TerraBind's affinity prediction performance across CASP16 and proprietary assay benchmarks, testing the hypothesis that latent pair representations from the structural Pairformer provide a sufficient signal for affinity without requiring all-atom 3D coordinates. By conditioning directly on pairwise distogram bin probabilities rather than structures generated via costly diffusion or coordinate optimization, TerraBind eliminates the all-atom refinement bottleneck. This approach is further supported by the observation that zero-shot distogram entropy ($H_{LP}$) correlates with binding affinity (Fig. 5). Integrated with a strong pretrained ligand encoding, this architecture exceeds Boltz-2 accuracy, demonstrating improved Pearson correlation across both CASP16 targets and a diverse proprietary dataset of 18 protein targets. The proprietary benchmark represents a rigorous test of out-of-distribution generalization,

as it comprises uncurated data from active drug discovery campaigns not seen during TerraBind's training on public datasets. Together, these results demonstrate that coarse structural latents are sufficient for state-of-the-art affinity prediction while utilizing a significantly leaner parameter footprint than current diffusion-based baselines.

### 3.2.2. STRUCTURAL FINE-TUNING IMPROVES AFFINITY

To demonstrate the value of structural information for affinity prediction, we isolated the effect of structural fine-tuning by performing a brief optimization of the Pairformer on a small proprietary crystallographic dataset. We evaluated the affinity module—without any retraining—using these updated representations. Even with minimal data (6 crystals for Target X and 3 crystals for Target Y), these targets with low sequence similarity ($< 20\%$) showed notable simultaneous improvements in Pearson correlation (Fig. 6). These results underscore the tight coupling between structural representation quality and affinity accuracy, highlighting how even sparse, program-specific structural data can enhance predictions for thousands of unseen molecules without even re-training the affinity module.

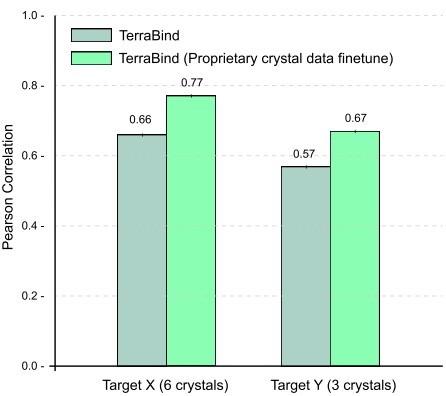

*Figure 6.* **Structure finetuning.** Finetuning on minimal crystal data enhances affinity accuracy on held-out molecules.

### 3.2.3. AFFINITY UNCERTAINTY QUANTIFICATION

Practical drug discovery requires well-calibrated uncertainty estimates to navigate "activity cliffs"—where subtle molecular changes produce dramatic potency differences (Passaro et al., 2025). We present the first uncertainty module for general co-folded binding models, leveraging the epinet to provide reliable error quantification.

Figure 7 illustrates this calibration by relating the interquartile range (IQR) of epinet samples to the "success rate", defined as the fraction of predictions within 1 $pIC_{50}$ (one log unit) of the true value. Across CASP16 and proprietary benchmarks, lower IQR (decreased uncertainty) strongly correlates with higher success rates. This relationship demonstrates that TerraBind's confidence estimates are well-calibrated, enabling researchers to prioritize high-certainty predictions in prospective screening campaigns.

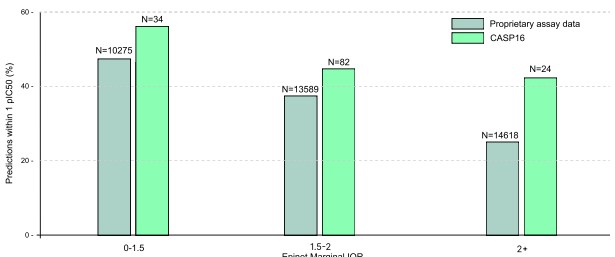

*Figure 7.* **Uncertainty-guided prioritization.** Higher success rates for low-IQR predictions validate the epinet as a robust tool for identifying high-certainty hits in prospective screening campaigns.

### 3.2.4. SIMULATED DMTA CYCLES WITH CONTINUAL LEARNING

Beyond marginal uncertainty, our architecture enables joint predictive distributions via shared epistemic indices (Section 2.4.2). As demonstrated in recent work (Wang-Henderson et al., 2025), this approach facilitates supe-

rior molecule selection over multiple Design-Make-Test-Analyze (DMTA) cycles and enables a *continual learning* scheme to incorporate experimental observations without retraining. We simulated a DMTA experiment using a pool of over 2,000 small molecules for Target I, a target characterized by harsh activity cliffs where subtle modifications lead to massive affinity shifts. In each cycle, a strategy selects 5 compounds for assay readout; these are removed from the pool and used to update the model (if applicable). Figure 8 tracks the regret—the difference between the global maximum $pIC_{50}$ in the pool and the best observed value up to that cycle.

We compared four selection strategies: *Boltz-2-greedy*, *TerraBind-greedy*, *Continual TerraBind-greedy*, and *Continual TerraBind-EMAX*. While base greedy approaches for both models showed similarly poor performance on this challenging target, the continual learning strategies quickly outperformed them. In particular, the EMAX acquisition function (Eq. 3) achieved the best results by effectively hedging selection risks and penalizing inferred correlations within a batch. Notably, these advanced selection and update schemes are uniquely enabled by the epinet-based likelihood module and cannot be readily applied to standard diffusion-based models like Boltz-2.

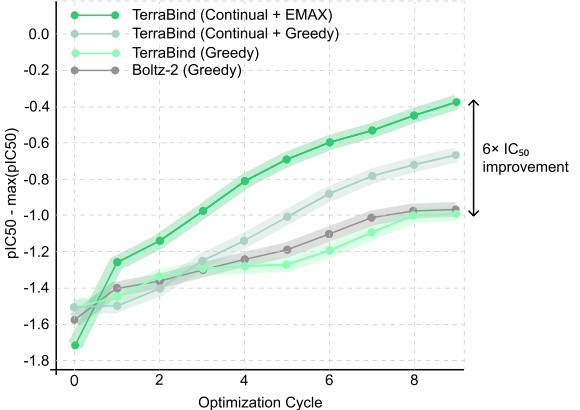

*Figure 8.* **Continual Learning Accelerates Hit Optimization.** Comparison of regret across selection strategies for Target I. Continual learning combined with the EMAX acquisition function enables faster discovery of top binders by leveraging joint posterior distributions and batch-aware risk hedging. Y-axis in log-scale.

## 4. Limitations

**Coarse-Grained Representations.** While TerraBind avoids all-atom refinement to achieve its $26\times$ speedup, the lack of atomic coordinates can limit downstream use in classical physics-based pipelines (Thaler et al., 2026). Furthermore, the optimization-based pose generation may struggle with convergence in very large protein systems where pairwise distograms become geometrically incompatible.

**Chemical Space and Context.** Structural models often generalize poorly to diverse ligand space even when protein sequences are well-represented. As noted in (Dobles et al., 2025), expanding ligand diversity via physics-based synthetic data remains a critical path for improvement. Additionally, our model considers only the immediate ligand-protein pair, which may not capture the complexity of cell-based assays or allosteric mechanisms where the broader biological context is required.

**Uncertainty Calibration.** The current epinet architecture yields Gaussian marginal distributions which can be overly optimistic, occasionally underestimating false-positive likelihoods. Future work involving prior network pretraining on non-Gaussian synthetic datasets could better mimic the skewed distributions seen in real-world assay data (Wang-Henderson et al., 2025).

## 5. Conclusion

We introduced TerraBind, a foundation model that resolves the computational bottlenecks of co-folded structures by demonstrating that all-atom diffusion is not a prerequisite for accurate binding affinity prediction. By leveraging rich molecular encoders with a streamlined Pairformer trunk, TerraBind achieves state-of-the-art performance with a $26\times$ increase in throughput over diffusion-based baselines like Boltz-2. Our results establish that coarse structural latents are sufficient for: (1) ligand pose accuracy competitive with all-atom diffusion models; (2) superior affinity prediction on CASP16 and proprietary benchmarks; and (3) well-calibrated uncertainty quantification. Furthermore, the model's modular design enables few-shot structural fine-tuning and advanced DMTA selection strategies via the EMAX acquisition function. TerraBind provides a scalable framework for deploying high-fidelity deep learning predictions across library-scale drug discovery campaigns and billion-scale proprietary datasets.

## Acknowledgements

We thank our colleagues at Terray Therapeutics for the experimental data, infrastructure, and scientific discussions that supported this work. We thank the NVIDIA cuEquivariance team for the optimized triangle attention and triangle multiplication kernels used throughout this paper. We also thank the maintainers of the open-source models and datasets that made this work possible, including the Protein Data Bank, BindingDB, ChEMBL, PubChem, and the teams behind ESM-2, AlphaFold, and Boltz.

## Impact Statement

This paper presents a foundation model for protein-ligand binding affinity prediction, with the goal of accelerating computational drug discovery. By enabling faster and more accurate compound screening, our work could reduce the time and cost of identifying therapeutic candidates, potentially broadening access to drug discovery capabilities for academic and resource-limited research groups.

As with any tool that facilitates molecular design, there exists a dual-use concern: the same methods that accelerate discovery of beneficial therapeutics could theoretically be applied to design harmful compounds. However, this risk is not unique to our approach and applies broadly to structure-based drug design methods. The coarse-grained representations and binding affinity focus of TerraBind do not provide capabilities beyond existing publicly available tools such as molecular docking software or other co-folding models.

We do not foresee specific negative societal consequences that must be highlighted beyond these standard considerations for computational chemistry research.

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

# A. Implementation Details

## A.1. Pairformer Architecture

The pairformer trunk follows the architecture introduced in Ref. (Abramson et al., 2024), with key modifications for computational efficiency. Each pairformer layer updates pair representations through two core operations: (1) *triangle attention*, and (2) *triangle multiplication*.

**Triangle attention** operates on pair representations by attending along rows or columns with biases derived from edges that complete triangles:

$$z'_{ij} = z_{ij} + \sum_k \text{softmax}_k \left( \frac{q_{ij} \cdot k_\alpha}{\sqrt{d_h}} + b_\beta \right) v_\alpha \quad \text{where} \quad (\alpha, \beta) = \begin{cases} (ik, jk) & \text{starting node} \\ (kj, ki) & \text{ending node} \end{cases} \tag{5}$$

where $q_{ij}, k_\alpha, v_\alpha$ are linear projections of the pair representations, $d_h$ is the head dimension, and $b_\beta$ is a learned bias derived from $z_\beta$—the edge completing the triangle $(i, j, k)$. The starting node variant attends along rows (fixed $i$), while the ending node variant attends along columns (fixed $j$).

**Triangle multiplication** propagates information through multiplicative gating over edges sharing a common node:

$$z'_{ij} = z_{ij} + \text{Linear} \left( \sum_k g_a(z_\gamma) \odot g_b(z_\delta) \right) \quad \text{where} \quad (\gamma, \delta) = \begin{cases} (ik, jk) & \text{outgoing} \\ (ki, kj) & \text{incoming} \end{cases} \tag{6}$$

where $g_a(\cdot)$ and $g_b(\cdot)$ are distinct gated linear projections and $\odot$ denotes element-wise multiplication. These operations encode geometric consistency: if residues $i$ and $j$ are both close to residue $k$, then $i$ and $j$ should be close to each other.

We use a 48-layer pairformer with 4-head attention and 128-dimensional pair representations. Unlike Ref. (Abramson et al., 2024), we omit single sequence representations entirely—these are only required for the downstream diffusion module, which is absent in TerraBind. This reduces the structure module parameter count from $\sim$147M to $\sim$27M.

We implement triangle attention and triangle multiplication using optimized CUDA kernels from NVIDIA cuEquivariance (v0.6.0) (NVIDIA, 2024), with all operations performed in bfloat16 precision. This provides $1.6\times$ training speedup and $3\times$ inference speedup compared to standard implementations while maintaining numerical stability. For full architectural details, we refer readers to Ref. (Abramson et al., 2024).

## A.2. Distance Binning and Structure Learning

### A.2.1. DISTANCE BIN CONFIGURATION

Following Ref. (Wohlwend et al., 2024), we project the final pair representations into $N_{\text{bins}} = 64$ pairwise distance bins. The bins are configured as follows: 62 bins evenly spaced from 2Å to 22Å (bin width $\approx 0.32$Å), plus two boundary bins for covalent-range distances ($< 2$Å) and long-range interactions ($> 22$Å). Distances are predicted between all ligand heavy atoms and protein residue centers (represented by $C_\beta$ atoms, or $C_\alpha$ for glycine). The pairformer produces a distogram of pairwise distance bin probability distributions, $p(d^b_{ij})$, where $i, j$ index atom pairs and $b \in \{1, \ldots, N_{\text{bins}}\}$ enumerates distance bins.

### A.2.2. STRUCTURE LEARNING LOSS

The distogram is trained via categorical cross-entropy over distance bins. For each pair of atoms $(i, j)$, we compute the pointwise loss:

$$\ell_{ij} = -\sum_{b=1}^{N_{\text{bins}}} y^b_{ij} \log p(d^b_{ij}) \tag{7}$$

where $y^b_{ij}$ is the one-hot encoding of the ground truth distance bin and $p(d^b_{ij}) = \text{softmax}(\mathbf{z}_{ij})_b$ is the predicted probability for bin $b$.

To emphasize binding-relevant geometry, we apply pair-type-specific weights. Let $\mathcal{M}$ denote the set of valid (non-padding,

non-diagonal) atom pairs. We construct a weight matrix $W$ where each entry $w_{ij}$ depends on the pair type:

$$w_{ij} = \begin{cases} w_{\text{LL}} & \text{if } i, j \in \text{ligand} \\ w_{\text{LP}} & \text{if } (i \in \text{ligand}, j \in \text{protein}) \text{ or } (i \in \text{protein}, j \in \text{ligand}) \\ w_{\text{PP}} & \text{if } i, j \in \text{protein} \end{cases} \tag{8}$$

The total structure loss is then computed as:

$$\mathcal{L}_{\text{struct}} = \frac{\sum_{(i,j) \in \mathcal{M}} w_{ij} \cdot \ell_{ij}}{\sum_{(i,j) \in \mathcal{M}} w_{ij}} \tag{9}$$

Pair types are distinguished as follows:

- **Ligand-ligand (LL)**: pairs where both atoms belong to ligand molecules, including both intra-ligand pairs (within the same molecule) and inter-ligand pairs (between different ligand molecules in multi-ligand systems)

- **Ligand-protein (LP)**: pairs between ligand atoms and protein residue centers

- **Protein-protein (PP)**: pairs where both atoms are protein residue centers, including both intra-chain pairs (within the same protein chain) and inter-chain pairs (between different chains in multi-chain complexes)

The weights $w_{\text{LL}}$, $w_{\text{LP}}$, and $w_{\text{PP}}$ are adjusted across training stages to progressively emphasize binding interface geometry (see Table 1). In Stage 1, all pair types are weighted equally ($w_{\text{LL}} = w_{\text{LP}} = w_{\text{PP}} = 1$). In Stage 2, we upweight ligand-ligand ($w_{\text{LL}} = 2$) and ligand-protein ($w_{\text{LP}} = 5$) pairs to focus learning on the binding site. Stage 3 returns to equal weights while training exclusively on experimental structures.

### A.2.3. EXPECTED PAIRWISE DISTANCE

From the predicted bin distributions, we compute expected pairwise distances as a probability-weighted average of bin centers:

$$\hat{d}_{ij} = \sum_{b=1}^{N_{\text{bins}}} p(d_{ij}^b) \cdot c_b \tag{10}$$

where $c_b$ is the midpoint of bin $b$. For boundary bins, we use $c_1 = 1.5\text{Å}$ (covalent) and $c_{64} = 24.5\text{Å}$ (long-range). These expected distances define the binding pocket:

$$\mathcal{P} = \{j \in \text{protein} : \exists i \in \text{ligand}, \hat{d}_{ij} < 15\text{Å}\} \tag{11}$$

This pocket definition serves as input context for both the affinity module (Section 2.4.1) and the pose module (Section 2.3).

### A.2.4. NORMALIZED PAIRWISE ENTROPY

We compute a normalized (information-theoretic) entropy for each pairwise distance distribution to quantify prediction confidence:

$$H(d_{ij}) = -\frac{1}{\log N_{\text{bins}}} \sum_{b=1}^{N_{\text{bins}}} p(d_{ij}^b) \log p(d_{ij}^b) \tag{12}$$

Normalization by $\log N_{\text{bins}}$ ensures $H(d_{ij}) \in [0, 1]$, where $H \approx 0$ indicates a peaked distribution (confident prediction) and $H \approx 1$ indicates a uniform distribution (uncertain prediction).

### A.2.5. AGGREGATED ENTROPY METRICS

We aggregate pairwise entropies by pair type to obtain summary confidence metrics. Let $L$ denote the set of ligand atoms and $\mathcal{P}$ the binding pocket residues (Eq. 11).

**Ligand-ligand entropy** averages over all ligand atom pairs:

$$H_{\text{LL}} = \frac{1}{|L|(|L|-1)} \sum_{\substack{i,j \in L \\ i \neq j}} H(d_{ij}) \tag{13}$$

**Ligand-protein entropy** averages over ligand-pocket interactions:

$$H_{\text{LP}} = \frac{1}{|L| \cdot |\mathcal{P}|} \sum_{i \in L} \sum_{j \in \mathcal{P}} H(d_{ij}) \tag{14}$$

**Protein-protein entropy** averages over pocket residue pairs:

$$H_{\text{PP}} = \frac{1}{|\mathcal{P}|(|\mathcal{P}|-1)} \sum_{\substack{i,j \in \mathcal{P} \\ i \neq j}} H(d_{ij}) \tag{15}$$

Note that $H_{\text{LL}}$ is computed over all ligand atoms, while $H_{\text{PP}}$ is restricted to pocket residues within the 15Å cutoff. The diagonal terms ($i = j$) are excluded from $H_{\text{LL}}$ and $H_{\text{PP}}$ as self-distances are trivially zero.

The ligand-protein entropy $H_{\text{LP}}$ serves as the primary confidence metric for structure prediction and provides a zero-shot binding affinity signal: lower $H_{\text{LP}}$ correlates with both higher pose accuracy and stronger binding (Section 3.2).

### A.3. Multi-Stage Training Protocol

We employ an effective batch size of 128 distributed across 4 NVIDIA H100 nodes and a 48-layer pairformer with 4-multihead pairwise attention. The TerraBind model was trained for a total of 105k steps following the multi-stage protocol described in Table 1.

*Table 1.* Multi-stage structure prediction training protocol. Data columns show sampling probabilities; loss weight columns show relative weighting for each pair type.

| Stage | Steps | Tokens | Data Sampling | | | Loss Weights | | |
|---|---|---|---|---|---|---|---|---|
| | | | PDB | AFDB | BindingDB | LL | LP | PP |
| 1 | 70k | 384 | 0.45 | 0.25 | 0.30 | 1× | 1× | 1× |
| 2 | 20k | 256 | 0.50 | 0.00 | 0.50 | 2× | 5× | 1× |
| 3 | 15k | 256 | 1.00 | 0.00 | 0.00 | 1× | 1× | 1× |

### A.4. Pocket Cropping Algorithm

The pocket cropping algorithm (Algorithm 1) enables efficient inference on large proteins by restricting computation to the binding site region. The algorithm operates in two stages: (1) an initial full-protein pairformer pass to identify pocket residues, and (2) a refined pass on the cropped context.

In the initial pass, pocket residues are identified using a 22Å cutoff on the expected pairwise distances $\hat{d}_{ij}$ (Eq. 10):

$$\mathcal{P}_{\text{init}} = \{j \in \text{protein} : \exists\, i \in \text{ligand}, \hat{d}_{ij} < 22\text{Å}\} \tag{16}$$

This larger cutoff (compared to the 15Å used for affinity prediction) ensures sufficient context for the refined pass.

The algorithm prioritizes including all ligand atoms and all pocket residues. If the combined token count exceeds the budget (rare), pocket residues are truncated by distance to ligand, keeping those closest. If the token budget permits additional context, the algorithm performs contiguous sequence expansion: it identifies clusters of pocket residues within each protein chain (merging residues separated by $\leq 3$ positions), then adds neighboring residues in order of sequence proximity to these clusters. This expansion strategy preserves local secondary structure context while respecting the computational budget.

---

**Algorithm 1** Pocket-Based Context Cropping

---

1: **Input:** Tokenized structure, budget $N_{\text{max}}$, pocket indices $\mathcal{P}_{\text{init}}$, expected distances $\{\hat{d}_{ij}\}$
2: **Output:** Cropped tokenized structure $\mathcal{C}$
3: // *Step 1: Always include entire ligand*
4: $\mathcal{L} \leftarrow \{i : \text{token}_i.\text{type} = \text{LIGAND}\}$
5: $\mathcal{C} \leftarrow \mathcal{L}$
6: // *Step 2: Include pocket residues (truncating if necessary)*
7: $\mathcal{P} \leftarrow \{i : \text{token}_i.\text{idx} \in \mathcal{P}_{\text{init}}\}$
8: **if** $|\mathcal{L}| + |\mathcal{P}| > N_{\text{max}}$ **then**
9:     $\text{budget}_{\text{pocket}} \leftarrow N_{\text{max}} - |\mathcal{L}|$
10:     Sort $j \in \mathcal{P}$ by expected distance to ligand $\min_{k \in \mathcal{L}} \hat{d}_{jk}$ (ascending)
11:     $\mathcal{P}_{\text{keep}} \leftarrow$ first $\text{budget}_{\text{pocket}}$ tokens from sorted $\mathcal{P}$
12:     $\mathcal{C} \leftarrow \mathcal{C} \cup \mathcal{P}_{\text{keep}}$
13: **else**
14:     $\mathcal{C} \leftarrow \mathcal{C} \cup \mathcal{P}$
15: **end if**
16: // *Step 3: Contiguous sequence expansion (if under budget)*
17: **if** $|\mathcal{C}| < N_{\text{max}}$ **then**
18:     candidates $\leftarrow []$
19:     **for** each protein chain $c$ intersecting $\mathcal{C}$ **do**
20:         $\mathcal{S}_c \leftarrow \{\text{res\_idx}(i) : i \in \mathcal{C} \cap \text{chain}_c\}$         ▷ pocket residues in chain
21:         Form clusters $\{(s_k, e_k)\}$ from $\mathcal{S}_c$ where residue gap $\leq 3$
22:         **for** each residue $r$ in chain $c$ not in $\mathcal{C}$ **do**
23:             $d_{\text{seq}} \leftarrow \min_k(\min(|r - s_k|, |r - e_k|))$         ▷ dist to cluster boundary
24:             candidates.append($(d_{\text{seq}}, r)$)
25:         **end for**
26:     **end for**
27:     Sort candidates by $d_{\text{seq}}$ (ascending)
28:     **for** $(d_{\text{seq}}, r) \in$ candidates **do**
29:         **if** $|\mathcal{C}| \geq N_{\text{max}}$ **then**
30:             **break**
31:         **end if**
32:         $\mathcal{C} \leftarrow \mathcal{C} \cup \{\text{token}(r)\}$
33:     **end for**
34: **end if**
35: // *Step 4: Finalize*
36: Filter structure to retain only tokens in $\mathcal{C}$ and induced bonds
37: **return** $\mathcal{C}$

---

### A.5. Pose Module

The pose module generates 3D coordinates from predicted distograms via gradient-based optimization (Algorithm 2). Given the pairformer's distance bin distributions, we first extract expected pairwise distances $\hat{d}_{ij}$ as probability-weighted averages of bin centers (Eq. 10). These expected distances define a target distance matrix that we use to optimize 3D coordinates via gradient descent on the joint ligand-pocket system.

We define the binding pocket $\mathcal{P}$ as protein residues within 15Å of any ligand atom based on predicted ligand-protein distances (Eq. 11). For the joint optimization over $N = L + |\mathcal{P}|$ points (ligand atoms plus pocket residues), we minimize the mean squared distance error:

$$\mathcal{L}_{\text{opt}} = \frac{1}{N(N-1)} \sum_{\substack{i,j \\ i \neq j}} w_{ij} \cdot \left( \|\mathbf{x}_i - \mathbf{x}_j\|_2 - \hat{d}_{ij} \right)^2 \tag{17}$$

where $\mathbf{x}_i \in \mathbb{R}^3$ are the coordinates being optimized and $\hat{d}_{ij}$ are the expected distances from the pairformer. All pair types (ligand-ligand, ligand-protein, and protein-protein) are weighted equally ($w_{ij} = 1$).

Coordinates are initialized from a standard normal distribution $\mathbf{x}_i \sim \mathcal{N}(0, \mathbf{I}_3)$ and optimized using Adam with learning rate $\alpha = 1.0$. Optimization terminates upon convergence (loss change $< 10^{-3}$ for 20 consecutive iterations) or after a maximum of 5000 iterations; in practice, convergence typically occurs within $\sim$500 iterations. After optimization, we perform rigid alignment with ground truth structures using equal weights for ligand and protein atoms.

---

**Algorithm 2** Joint Ligand-Pocket Coordinate Optimization

---

1: **Input:** Pairformer logits $\mathbf{Z}^{LL} \in \mathbb{R}^{L \times L \times B}$, $\mathbf{Z}^{PP} \in \mathbb{R}^{P \times P \times B}$, $\mathbf{Z}^{LP} \in \mathbb{R}^{L \times P \times B}$
2: **Input:** Bin centers $\{c_b\}_{b=1}^{B}$, pocket cutoff $d_{\text{cut}} = 15\text{Å}$
3: **Output:** Optimized coords $\mathbf{X}_{\text{lig}}$, $\mathbf{X}_{\text{pocket}}$, and pocket indices $\mathcal{P}$
4:                                                                              ▷ $L$ = ligand heavy atoms, $P$ = protein C$\beta$ atoms, $B$ = distance bins
5:
6: // *Step 1: Compute expected distance matrices from logits*
7: **for** each pair type $t \in \{\text{LL}, \text{PP}, \text{LP}\}$ **do**
8:    $p_{ij}^{b} \leftarrow \text{softmax}(\mathbf{Z}_{ij}^{t})_b$                                     ▷ bin probabilities
9:    $\hat{D}_{ij}^{t} \leftarrow \sum_{b=1}^{B} p_{ij}^{b} \cdot c_b$                              ▷ expected distance (Eq. 10)
10:    $\hat{\mathbf{D}}^{t} \leftarrow [\hat{D}_{ij}^{t}]$                                           ▷ expected distance matrix
11: **end for**
12:
13: // *Step 2: Identify pocket residues*
14: $\mathcal{P} \leftarrow \{j \in [P] : \min_{i \in [L]} \hat{D}_{ij}^{LP} < d_{\text{cut}}\}$
15: $K \leftarrow |\mathcal{P}|, \quad N \leftarrow L + K$
16:
17: // *Step 3: Build reference distance matrix*
18: $\mathbf{R} \leftarrow \mathbf{0}^{N \times N}$
19: $\mathbf{R}_{[1:L, \, 1:L]} \leftarrow \hat{\mathbf{D}}^{LL}$                                     ▷ ligand-ligand block
20: $\mathbf{R}_{[L+1:N, \, L+1:N]} \leftarrow \hat{\mathbf{D}}^{PP}[\mathcal{P}, \mathcal{P}]$        ▷ pocket-pocket block
21: $\mathbf{R}_{[1:L, \, L+1:N]} \leftarrow \hat{\mathbf{D}}^{LP}[:, \mathcal{P}]$                    ▷ ligand-pocket block
22: $\mathbf{R}_{[L+1:N, \, 1:L]} \leftarrow \mathbf{R}_{[1:L, \, L+1:N]}^{\top}$                      ▷ symmetry
23:
24: // *Step 4: Optimize coordinates via gradient descent*
25: $\mathbf{X} \sim \mathcal{N}(\mathbf{0}, \mathbf{I}), \quad \mathbf{X} \in \mathbb{R}^{N \times 3}$
26: Initialize Adam optimizer with learning rate $\alpha = 1.0$
27: stable $\leftarrow 0$
28: **for** $t = 1$ to $T_{\text{max}} = 5000$ **do**
29:    $D_{ij} \leftarrow \|\mathbf{x}_i - \mathbf{x}_j\|_2$                                        ▷ current pairwise distances
30:    $\mathcal{L} \leftarrow \frac{1}{N(N-1)} \sum_{i \neq j} (D_{ij} - R_{ij})^2$
31:    $\mathbf{X} \leftarrow \text{Adam\_step}(\mathbf{X}, \nabla_{\mathbf{X}} \mathcal{L})$
32:    **if** $|\mathcal{L}^{(t)} - \mathcal{L}^{(t-1)}| < 10^{-3}$ **then**
33:       stable $\leftarrow$ stable $+ 1$
34:       **if** stable $\geq 20$ **then**
35:          **break**
36:       **end if**
37:    **else**
38:       stable $\leftarrow 0$
39:    **end if**
40: **end for**
41:
42: **return** $\mathbf{X}_{[1:L]}$, $\mathbf{X}_{[L+1:N]}$, $\mathcal{P}$

---

## A.6. Continual learning scheme

Here we employ a pathwise conditioning approach rooted in Gaussian Process (GP) regression (Matheron, 1973; Wilson et al., 2020). We treat the ensemble of predictions from the Epinet as samples from an implied prior stochastic process $f \sim \mathcal{GP}(\mu, K)$. Instead of defining an explicit kernel function, we utilize the empirical covariance of the ensemble outputs to approximate the kernel $K$. Given a set of observed data points $\mathcal{D}_{obs} = \{(\mathbf{x}_i, y_i)\}_{i=1}^{N}$ and a set of unobserved query points $\mathcal{X}_*$, we update the ensemble predictions directly using the linear update rule for conditional Gaussian simulations. The updated prediction vector $\hat{\mathbf{y}}_{new}$ at the query points is computed as:

$$\hat{\mathbf{y}}_{new} = \hat{\mathbf{y}}_{prior} + \hat{K}_* (\hat{K} + \sigma_{obs}^2 I)^{-1} (\mathbf{y}_{true} - \hat{\mathbf{y}}_{obs} - \boldsymbol{\epsilon}) \tag{18}$$

where $\hat{\mathbf{y}}_{prior}$ denotes the original Epinet predictions at $\mathcal{X}_*$, $\hat{\mathbf{y}}_{obs}$ denotes predictions at the observed locations, and $\mathbf{y}_{true}$ is the vector of ground truth observations. The matrices $\hat{K}$ and $\hat{K}_*$ represent the empirical covariance between observed points, and between unobserved and observed points, respectively. We also inject independent Gaussian noise $\boldsymbol{\epsilon} \sim \mathcal{N}(0, \sigma_{obs}^2 I)$,

with $\sigma_{obs} = 0.5$, into the residual term for each sample path (Wilson et al., 2020; Garnelo et al., 2018). This simple heuristic allows for rapid Bayesian conditioning on new observations without the expense of retraining.

### A.7. Structural Fine-tuning

To fine-tune our structural model on Proprietary crystal structures, we use the standard structural model training protocol with the following modifications: a smaller learning rate ($10^{-5}$), shorter training duration (5k steps), and a low proprietary crystal sampling rate ($0.01\%$) relative to experimental PDB structures. We observe that more aggressive fine-tuning (higher learning rates or sampling rates) tends to yield models that overfit on internal data and generalize poorly on public structure benchmarks.

### A.8. Boltz Inference Details

Unless otherwise specified, Boltz-1 and Boltz-2 inferences were performed with default inference parameters. For timing comparisons with TerraBind, we use the same NVIDIA cuEquivariance (v0.6.0) package (NVIDIA, 2024), mixed precision (bfloat16), and hardware (single A6000 GPU) to ensure fair comparison.

## B. Results details

### B.1. Runtime Decomposition and Efficiency Analysis

In Figure 9, we present a detailed latency decomposition comparing TerraBind against Boltz-2, benchmarked on a single NVIDIA A6000 GPU for a protein-ligand complex of 196 tokens. Note that these runtime metrics exclude Multiple Sequence Alignment (MSA) generation. The reported latencies represent an end-to-end structure and potency inference, generating 10 pose samples and a binding affinity prediction. The observed $26.6\times$ aggregate speedup is primarily driven by the elimination of the iterative diffusion denoising process. As shown in the "Pose" component, Boltz-2 requires $25.92$ seconds for coordinate generation. In contrast, our batched pose optimization requires only $0.17$ seconds, representing a reduction of over two orders of magnitude. Furthermore, the "Trunk" latency is reduced from $1.53$ seconds (Boltz-2) to $0.87$ seconds (TerraBind). This $1.75\times$ acceleration stems from our streamlined architecture, which utilizes a 48-layer Pairformer (compared to 64 layers in Boltz-2) and eliminates the single sequence representation track. Notably, the reported TerraBind trunk latency includes the computational cost of the pre-trained encoders (ESM-2 for protein and the COATI-3 molecular encoder), yet still significantly outperforms the baseline. Additionally, unlike Boltz-2, TerraBind does not employ a separate confidence module; the auxiliary head for affinity prediction incurs negligible overhead ($0.06$ seconds), validating our design choice to perform potency tasks directly on latent structural embeddings without requiring full coordinate reconstruction. In a "minimal affinity setup"—defined as TerraBind without pose generation versus Boltz-2 restricted to 5 diffusion samples—we observe a $17.01\times$ effective speedup for a 196 token context.

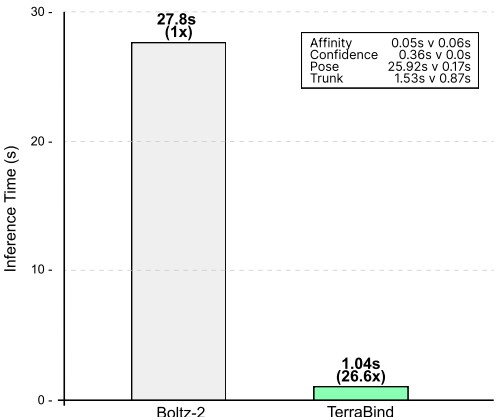

*Figure 9.* **Inference Latency Decomposition.** Comparison of end-to-end runtime (Structure + Potency) between Boltz-2 and TerraBind when generating 10 pose samples. The replacement of iterative diffusion with batched pose optimization accounts for the majority of the $26.6\times$ speedup. Additionally, the trunk is accelerated by using a shallower 48-layer Pairformer without a single sequence representation.

## B.2. PDB-Only Fine-Tuning (Stage 3)

Our multi-stage training protocol concludes with Stage 3: fine-tuning exclusively on experimental PDB structures with equal loss weights. Figure 10 shows the impact of the three training stages on distogram confidence (heldout validation set).

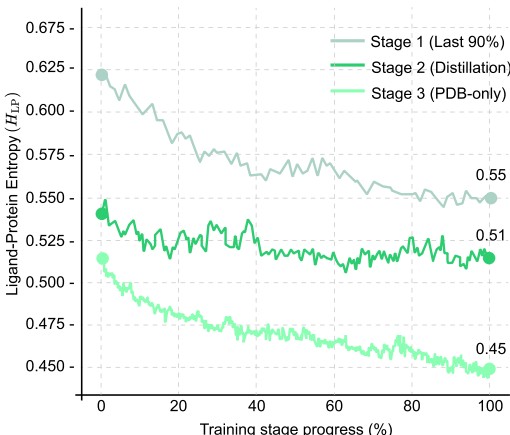

*Figure 10.* **Impact of Stage 3 PDB-only fine-tuning on distogram entropy.** Ligand-protein entropy over the three training stages. Stage 3 substantially reduces entropy, indicating more confident geometric predictions when trained on highest-quality experimental data.

Stage 3 fine-tuning shifts the entropy distribution toward lower values, indicating the model becomes more confident about binding geometry when trained exclusively on high-quality experimental structures. This entropy reduction occurs without degrading RMSD performance, suggesting the model learns to place probability mass more decisively on the correct binding mode rather than hedging across alternatives. We hypothesize that distillation data (AFDB, BindingDB) in earlier stages, while valuable for diversity, introduces some geometric ambiguity that is resolved by final fine-tuning on ground truth structures.

## B.3. Protein Encoder Ablations

To evaluate the impact of protein encoder choice on structure prediction, we trained model variants with different protein encoders using the same training regime as TerraBind. The pretrained ligand encoder remained fixed across experiments. We compare ESM-2 (650M), E1 (600M) (Jain et al., 2025), and MSA-based encoding. For the MSA baseline, we use the representations from Boltz-1 Trunk and apply our coordinate optimization for pose generation. Figure 11 reports protein RMSD < 2Å success rate and protein-protein entropy ($H_{PP}$) aggregated across FoldBench, PoseBusters, and Runs N' Poses benchmarks. Notably, E1 outperforms both ESM-2 and MSA-based encoding, achieving the highest success rate and lowest entropy.

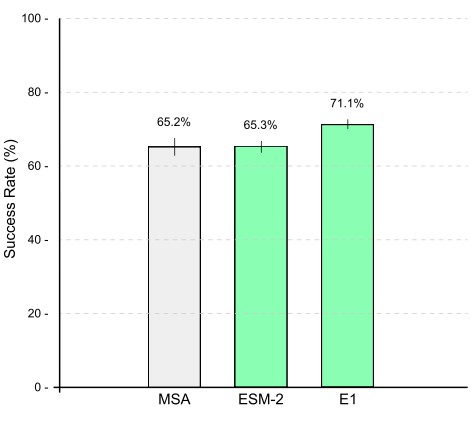

*(a)* Protein RMSD < 2Å success rate.

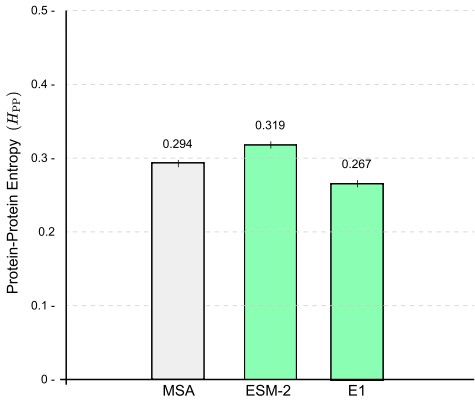

*(b)* Protein-protein entropy $H_{PP}$.

*Figure 11.* **Protein encoder ablation.** (a) Protein RMSD < 2Å success rate and (b) protein-protein entropy $H_{PP}$ aggregated across FoldBench, PoseBusters, and Runs N' Poses for TerraBind (ESM-2, 650M), TerraBind-E1 (Profluent-E1, 600M), and MSA via Boltz-1 Trunk. TerraBind-E1 achieves the highest protein RMSD success rate and the lowest $H_{PP}$, indicating more confident and accurate protein structure predictions.

## B.4. Affinity Data Distributions

To highlight the difficulty associated with real-world drug discovery campaign data, we quantify experimental assay data using the following activity cliff metrics: for molecule pairs of high similarity (Tanimoto similarity > 0.7) we obtain the *activity cliff IQR*: interquartile range (IQR) of the pairwise $|\Delta pIC_{50}|$ and the *activity cliff rate*: fraction of pairs exhibiting activity cliffs ($|\Delta pIC_{50}| > 1$). These metrics are computed per assay and then averaged across the assays within the dataset. We compare these metrics across different datasets in Table 2, including the Merck/JACS set (Wang et al., 2015) where only assays with > 20 compounds are retained. Notably, we find that our proprietary data is considerably more challenging than public datasets in terms of IQR and activity cliff rates. We use Merck/JACS only to characterize difficulty, not as a predictive benchmark, since we do not follow the same training target holdouts used in Boltz-2 (Passaro et al., 2025).

For target I, the simulated DMTA experiment is an especially challenging task for models. The distribution of potency is shown in Table 3, which contains a high frequency of low potency compounds in the compound selection pool.

*Table 2.* Comparison of IQR and Activity Cliff Rate across different datasets.

| Dataset / Target Set | IQR | Activity Cliff Rate (%) |
|---|---|---|
| Combined Merck/JACS sets (> 20 compound assays) | 0.63 | 18 |
| CASP16 | 0.51 | 18 |
| Proprietary targets | 0.82 | 25 |

*Table 3.* Experimental assay potency distribution for Target I (simulated DMTA experiment).

| Potency Range ($pIC_{50}$) | Compound Count |
|---|---|
| $pIC_{50} < 6$ | 1,238 |
| $6 \leq pIC_{50} < 7$ | 373 |
| $7 \leq pIC_{50} \leq 8$ | 389 |
| $pIC_{50} > 8$ | 22 |
| Total Compounds | 2,022 |

## B.5. Limited Binding Site Context for Affinity Inference

In Figure 12, we evaluate affinity predictions for the full set of CASP16 L3000 complexes. We adopt a "minimal token budget" strategy by defining a single binding site context derived from the largest ligand in the set. Specifically, we perform one full-sequence structural prediction for this ligand and identify protein residues with expected distances less than 15Å from predicted ligand atoms. This process yielded a 128-residue context, which was subsequently used as input for structure and affinity inference across all molecules. This approach relies entirely on predicted structures. Despite reducing the input from the full 846-residue sequence to just 128 tokens, we observe a $0.95+$ Pearson R correlation with full-context predictions. Benchmarking on a single NVIDIA A6000 GPU demonstrates a $20.3\times$ speedup for affinity prediction ($17.16\times$ with pose generation) while maintaining roughly equivalent accuracy.

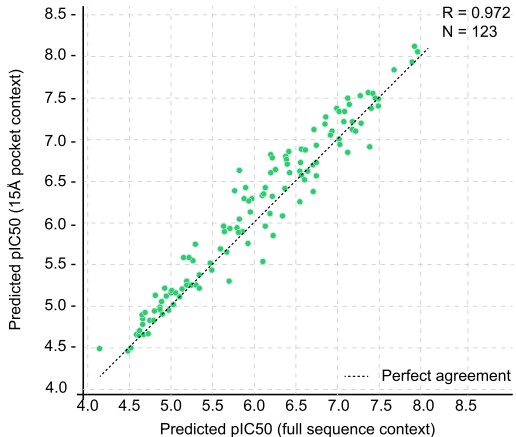

*Figure 12.* **Efficiency of limited binding site context on CASP16.** Performance comparison on the full L3000 dataset. The binding site context was defined via the predicted structure of the largest ligand, resulting in a 128-token pocket subset applied to all complexes. This minimal context achieves a $0.95+$ Pearson R correlation with full-sequence results and delivers a $17.16\times$ to $20.3\times$ speedup (single A6000 GPU) relative to the full 846-residue baseline.

## B.6. Sequence-based model baseline

In Fig. 13, we plot the performance of a straightforward *sequence-based* model architecture to serve as a model baseline. This 5M parameter model is constructed by a series of MLPs. Two input MLPs are used to independently process the end-of-sequence ESM-2 residue embedding for the sequence and the ligand embedding. The output embeddings are concatenated and provided to another MLP before diverging at the quantitative and binary output prediction heads. All other aspects of the training data and training process are identical to the process described in Section 2.4.3.

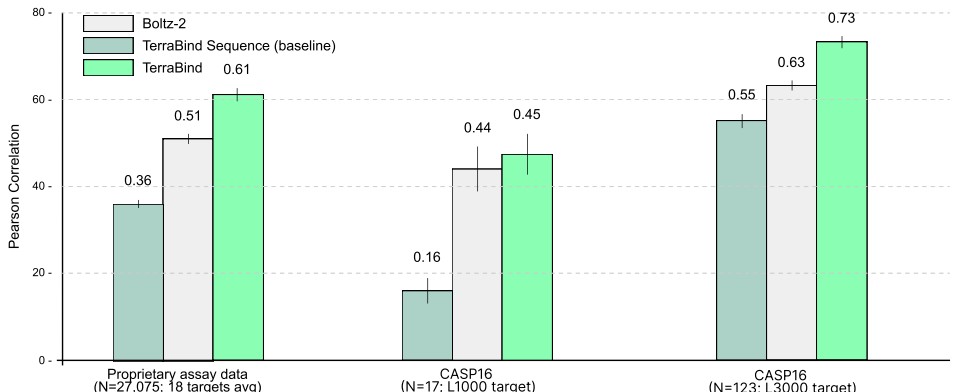

*Figure 13.* **Sequence-based model baseline.** A 5M parameter MLP architecture using ESM-2 and the same aggregate SMILES ligand embedding as TerraBind without structural information. TerraBind's structure-aware approach substantially outperforms this baseline.

## B.7. Binding Pocket Context Size Distribution

Figure 14 shows the distribution of binding pocket context sizes across the affinity training dataset. For each protein-ligand complex, the pocket context is defined as the number of ligand heavy atoms plus protein residues predicted to be within 15Å of any ligand atom, based on expected distances from the pairformer's distogram (Eq. 10).

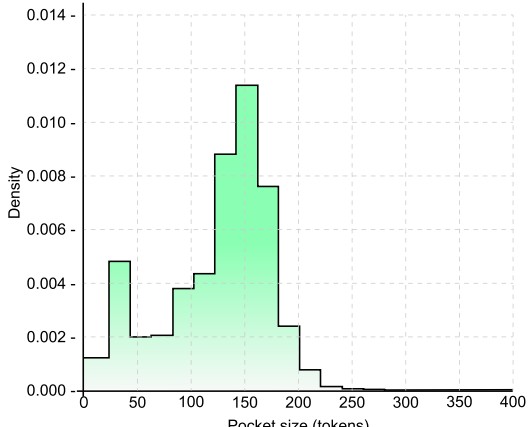

*Figure 14.* **Binding pocket context size distribution.** Histogram of token count ($N_{\text{token}}$ = ligand heavy atoms + pocket residues within 15Å) aggregated over the binding affinity dataset. The distribution peaks around 100–150 tokens, validating our 256-token crop size for training and 196-token inference context for TerraBind Pocket.

The distribution peaks around 150 tokens, with the vast majority of small-molecule binding contexts requiring fewer than 200 tokens. This empirical observation validates our architectural choices: the 256-token crop size used in Stages 2 and 3 of training, and the 196-token inference context for TerraBind Pocket, are sufficient to capture the full binding site for nearly all drug-like protein-ligand complexes.

## B.8. Context-Size Ablation

To assess sensitivity to the inference context size, we co-fold each complex with the full protein sequence, then crop to a fixed token budget around the predicted pocket and re-fold on Runs N' Poses ($n = 2,687$). Pose accuracy is stable across common crop sizes (Table 4), confirming that binding-relevant geometry is concentrated in the pocket and that the 196-token inference context used by TerraBind Pocket incurs negligible accuracy loss relative to full-sequence inference.

*Table 4.* Context-size ablation on Runs N' Poses. Cropping the protein context to a fixed token budget has negligible effect on pose accuracy.

| Context | Lig. RMSD $< 2$Å (%) | RMSD $< 2$Å & LDDT-PLI $> 0.8$ (%) |
|---|---|---|
| Full sequence | 61.6 | 49.3 |
| 256 tokens | 61.2 | 48.8 |
| 196 tokens | 61.4 | 49.0 |
| 164 tokens | 61.4 | 49.0 |

## B.9. Distance-Bin Ablation

To assess sensitivity to the distance-bin discretization, we trained a variant with a log-spaced binning scheme (1.2–50.75 Å) in place of the default linear scheme (2–22 Å). Pose accuracy is comparable across the two configurations on FoldBench (Table 5), indicating that TerraBind is robust to this design choice. We adopt the 2–22 Å linear scheme as the default. As with the context-size ablation, these are directional sensitivity checks rather than fully optimized comparisons.

*Table 5.* Distance-bin ablation on FoldBench. Pose accuracy is comparable between the default linear binning (2–22 Å) and a log-spaced alternative (1.2–50.75 Å).

| Bin configuration | Lig. RMSD $< 2$Å (%) | Prot. RMSD $< 2$Å (%) | LDDT-PLI | $H_{LP}$ |
|---|---|---|---|---|
| 2–22 Å linear (TerraBind) | 52.0 | 65.8 | 0.599 | 0.470 |
| 1.2–50.75 Å log-space | 50.0 | 71.6 | 0.581 | 0.439 |

## C. Evaluating TerraBind and Boltz-2 Across Hit-Finding and Hit-to-Lead Regimes

The main text reports Pearson correlations across CASP16 and on 18 proprietary biochemical assays (Section 3.2). This appendix extends that evaluation in two directions. On the 18 proprietary assays, we report results under two additional regimes: *hit-finding*, which evaluates binary discrimination of binders from inactives within each assay (AUROC at $pIC_{50} > 5$), and *hit-to-lead*, which evaluates ranking among confirmed binders (Pearson, Spearman, and Kendall correlations on the subset with $pIC_{50} > 5$). On CASP16, we extend the comparison to Spearman and Kendall.

This regime-aware evaluation also makes use of an additional Boltz-2 output. Alongside its quantitative affinity head, Boltz-2 provides a binary binding head; the two are combined into a score that is relevant when the evaluated set may contain inactives. Reporting both Boltz-2 outputs alongside TerraBind's single quantitative head—which is the same output used across all regimes—clarifies how each model's design choices map onto the different evaluation regimes.

### C.1. Model Variants

We compare three affinity outputs across this section:

- **TerraBind Quant:** TerraBind's quantitative head, returning a calibrated $pIC_{50}$.
- **Boltz-2 Affinity:** Boltz-2's quantitative head $\hat{a}$, reported as $\log_{10}(IC_{50}/\mu M)$ and converted to a calibrated $pIC_{50}$ via $pIC_{50} = 6 - \hat{a}$.
- **Boltz-2 Combined:** A score, proposed in the original work for virtual screening purposes (Passaro et al., 2025), that fuses Boltz-2's affinity head with its binary binding head $p^{bind} \in [0, 1]$. The need for such a score stems from the fact that the Boltz-2 quantitative head is trained on confirmed binders only, rendering it a model for binding affinity conditional on the evaluated compound being a true binder. This becomes problematic whenever the evaluated set may contain inactives, where binders and non-binders must be distinguished while simultaneously selecting the most potent binders. The combined score therefore prioritizes high-affinity compounds via the quantitative head while ensuring the compound has high predicted binding probability under the binary head:

$$\text{score}_{\text{combined}} := \max\left(\frac{-\hat{a} + 2}{4}, \ 0\right) \cdot p^{bind}. \tag{19}$$

The score lies on an arbitrary non-negative scale with no direct $pIC_{50}$ interpretation.

### C.2. Scenario 1: Hit-Finding (AUROC at $pIC_{50} > 5$)

In the hit-finding regime, AUROC measures how well each model separates binders ($pIC_{50} > 5$, $10\,\mu M$) from non-binders within each assay. Because AUROC is scale-free, it admits a fair comparison against TerraBind Quant for both Boltz-2 outputs—the calibrated affinity head and the arbitrary-scale combined score. Since the metric requires both classes, the $\geq 50$ and $\geq 100$ filters apply *per class* (binders *and* non-binders)—a stricter, two-sided requirement than the binder-only filter in the ranking scenarios (Section C.3); the assays here are a subset of those evaluated in hit-to-lead.

*Table 6.* Hit-finding AUROC at $pIC_{50} > 5$. Boltz-2 cells report the AUROC value, percent difference from TerraBind Quant, and per-target win count for TerraBind Quant.

| Threshold (assays) | TerraBind Quant | Boltz-2 Affinity | Boltz-2 Combined |
|---|---|---|---|
| $\geq 50$ / target (13) | 0.802 | 0.765 (+4.9%, 11/13) | 0.835 (−3.9%, 2/13) |
| $\geq 100$ / target (10) | 0.819 | 0.779 (+5.2%, 9/10) | 0.857 (−4.4%, 1/10) |

TerraBind Quant leads against Boltz-2 Affinity on the calibrated comparison, while Boltz-2 Combined leads against TerraBind Quant on the AUROC comparison. The combined score draws additional classification power from a binary binding head trained on PubChem HTS data ($\sim$2M compounds across $\sim$300 targets) not used by TerraBind (Passaro et al., 2025). The score itself, however, does not return a calibrated $pIC_{50}$: hit-finding selection with the combined score is restricted to relative top-$k$ ranking, since a fixed potency cutoff ("advance everything predicted above 10 µM") has no corresponding operating point on its scale.

### C.3. Scenario 2: Hit-to-Lead (Ranking at $pIC_{50} > 5$)

For ranking potency among confirmed binders ($pIC_{50} > 5$), there are no inactives to discriminate, so the affinity head applies directly and the comparison reduces to TerraBind Quant and Boltz-2 Affinity. We report Pearson, Spearman, and Kendall correlations. Restricting to binders reduces effective $N$ per target; we filter to assays with $N \geq 50$ and $N \geq 100$ binders to prevent low-$N$ targets from dominating the across-target average.

*Table 7.* Hit-to-lead ranking at $pIC_{50} > 5$. Each cell shows TerraBind Quant / Boltz-2 Affinity, with relative gain and per-target win count for TerraBind Quant.

| Threshold (assays) | Pearson | Spearman | Kendall |
|---|---|---|---|
| $N \geq 50$ (14) | 0.470 / 0.491 ($-4.3\%$, 8/14) | 0.425 / 0.409 ($+4.0\%$, 7/14) | 0.298 / 0.288 ($+3.6\%$, 7/14) |
| $N \geq 100$ (10) | 0.536 / 0.486 ($+10.2\%$, 8/10) | 0.477 / 0.396 ($+20.4\%$, 7/10) | 0.337 / 0.275 ($+22.5\%$, 7/10) |

### C.4. CASP16

CASP16 contains only confirmed binders with measured affinity—there are no inactive compounds to discriminate—so the regime split from the proprietary assays does not apply. The comparison reduces to the two calibrated heads: TerraBind Quant and Boltz-2 Affinity.

*Table 8.* CASP16 correlations on L3000 ($N = 123$) and L1000 ($N = 17$). Each Boltz-2 cell shows the correlation value and relative gain for TerraBind Quant.

| Target | Metric | TerraBind Quant | Boltz-2 Affinity |
|---|---|---|---|
| L3000 ($N = 123$) | Pearson | 0.725 | 0.625 ($+16.0\%$) |
| | Spearman | 0.722 | 0.626 ($+15.3\%$) |
| | Kendall | 0.544 | 0.451 ($+20.6\%$) |
| L1000 ($N = 17$) | Pearson | 0.470 | 0.470 ($0.0\%$) |
| | Spearman | 0.355 | 0.311 ($+14.1\%$) |
| | Kendall | 0.235 | 0.206 ($+14.1\%$) |

### C.5. Discussion

The 18 proprietary assays span both early-stage chemically diverse screens and analog series from active programs. As a complementary view, we partition them using internal campaign metadata rather than $pIC_{50}$ thresholds: compounds assigned to a named chemical series are labeled hit-to-lead ($\sim$14k compounds), the rest hit-finding ($\sim$13k compounds). Both subsets retain a substantial inactive fraction ($pIC_{50} < 4$), though the hit-to-lead subset has a larger fraction in the $pIC_{50}$ 5–8 range as expected from targeted series optimization. This distribution differs from curated benchmarks typically used to evaluate ranking, where assays are pre-filtered to confirmed binders within a small potency window. The partition illustrates a basic fact about real campaign data: the same assay typically contains both regimes' compounds, and no clean a priori labeling separates them—any deployed model has to handle both.

This is where the architectural difference between the two models matters. TerraBind Quant is trained on both active and inactive compounds within a single regression objective, so a single output discriminates binders from inactives *and* ranks potency among binders. Boltz-2 Affinity is trained on quantitative data from confirmed binders and relies on a separate binary head for classification; the binary head additionally draws on PubChem HTS data ($\sim$2M compounds across $\sim$300 targets) (Passaro et al., 2025), and the two heads are combined in the score of Eq. 19. This two-output design carries two practical limitations for deployment. First, the combined score lies on an arbitrary scale: it supports only relative top-$k$ selection ("advance the top 10%"), while a fixed $pIC_{50}$ cutoff ("advance everything predicted above 10 µM")—the

standard currency of medicinal chemistry—is unavailable. Second, the two outputs target different regimes: the combined score incorporates the binary head to suppress inactives and is the output Boltz-2 uses for screening, while the affinity head—trained on confirmed binders—is the appropriate output for ranking potency among them. The practitioner therefore has to know which regime an assay belongs to *a priori* to pick the right output—a determination that, as the metadata partition shows, is not always separable from the data itself. Consistent with these constraints, recent applications of Boltz-2 to affinity prediction tend to use the affinity head alone (Kim et al., 2025; Wan et al., 2026).

TerraBind's single calibrated $pIC_{50}$ output spans hit-finding, hit-to-lead, and full-population evaluation without regime-dependent output selection, on an interpretable scale (a prediction of 6 communicates an expected $IC_{50}$ of 1 μM). This makes a single deployment workable across the full breadth of a drug-discovery campaign—from diverse-library triage through analog ranking—with no head swapping, score combination, or per-stage calibration.

