# OpenReview forum: "TerraBind: Fast and Accurate Binding Affinity Prediction through Coarse Structural Representations"
_ICML.cc/2026/Conference — ICML 2026 regular_

### Official Review · Reviewer_kb5F · 2026-03-11

**Soundness:** 4
**Presentation:** 4
**Significance:** 4
**Originality:** 3
**Overall Recommendation:** 5
**Confidence:** 3

**Summary:**

In this paper, the authors propose CoarseBind, which uses a coarse pocket-level representation that includes only protein $C_\beta$ atoms and ligand heavy atoms. The key innovation of the mode is to use a coarse representation to replace the all atom representations and use a diffusion-free optimization module to replace widely-used diffusion models. On the benchmark datasets, the proposed model can achieve great accuracy and outperforms Boltz-2 by a large margin in binding affinity prediction.

**Compliance With Llm Reviewing Policy:**

Affirmed.

**Key Questions For Authors:**

I am not an expert in this area, but the results and approaches sound very convincing!

One simple concern is that the authors use only $C_\beta$ to represent protein structure, in this way, it may lose lot of the information about the amino acid side chain. Some important physics may be missing, such as hydrogen-bonding, \pi-\pi stacking, etc.

**Limitations:**

N.A.

**Strengths And Weaknesses:**

Strengths: a novel coarse representation and diffusion-free optimization process.

Weaknesses: coarse representation may lose critical physics such as hydrogen-bonding, \pi-\pi stacking, etc.

---

> ### Author Rebuttal · Authors · 2026-03-30
>
> We thank the reviewer for the positive feedback.
>
> > One simple concern is that the authors use only  to represent protein structure, in this way, it may lose lot of the information about the amino acid side chain. Some important physics may be missing, such as hydrogen-bonding, \pi-\pi stacking, etc.
>
> This is a correct observation. Interestingly, our empirical results suggest that even though these interactions are physically important, explicitly modeling them at the all-atom level is not necessary to achieve accurate structure and affinity predictions. Boltz-1/2 operate at full atomic resolution, yet CoarseBind matches or exceeds its performance across the benchmarks we present. Hence, we presume that the model has learned these interactions implicitly at the coarse resolution.

---

> > ### Author Rebuttal · Reviewer_kb5F · 2026-04-05
> >
> > I have no further comments and will keep my score!

---

### Official Review · Reviewer_uUyB · 2026-03-11

**Soundness:** 3
**Presentation:** 3
**Significance:** 3
**Originality:** 3
**Overall Recommendation:** 5
**Confidence:** 4

**Summary:**

the authors propose coarsebind, a method for predicting protein-ligand structures and binding affinities. It also challenges the necessity of full-atom or heavy diffusion architectures. Benchmark results demonstrate its performance.

**Compliance With Llm Reviewing Policy:**

Affirmed.

**Key Questions For Authors:**

* Generality Assessment: How sensitive are the results to key parameters, such as distance bins, pocket cropping, or validation splits of proteins? You are using much smaller models, so I believe deeper generalization studies are needed to evaluate potential overfitting or data leakages.
* For continual learning, how does the pathwise GP approximation scale with observation count?
* necessary and effiency of epinet,  such as in many HTVS tailed distribution settings (activity cliff )?

**Limitations:**

above

**Strengths And Weaknesses:**

Cons:
* The 26× speedup over Boltz-2 is compelling, this makes CoarseBind practical for HTVS.
* Epinet delivers calibrated uncertainties, enabling hedged batch selection in DMTA simulations, and continual learning via pathwise conditioning is innovative and efficient, avoiding full retraining.
* overall, the framework is very practical and i think should be helpful in drug discovery pipelines, including lead optimization, htvs etc.

Pros:
* the reliance on predicted pockets could falter in cases of extreme induced fit docking or allosteric binding, where broader context is crucial

---

> ### Author Rebuttal · Authors · 2026-03-30
>
> We thank the reviewer for the thorough review.
>
> > the reliance on predicted pockets could falter in cases of extreme induced fit docking or allosteric binding, where broader context is crucial
>
> We agree this is a valid concern. We note that the CoarseBind affinity module is trained and evaluated on the full sequence rather than pocket crops. The pocket-level *structure* metrics are presented only to demonstrate that the model retains accurate structural information at the binding site. Accordingly, CoarseBind should be more robust to induced-fit and allosteric effects compared to Boltz-2, whose affinity module is trained on pocket crops. We will clarify this distinction in the revised paper.
>
> > Generality Assessment: How sensitive are the results to key parameters, such as distance bins, pocket cropping, or validation splits of proteins?
>
> We have conducted additional experiments demonstrating robustness to key hyperparameters. These are preliminary ablations intended to assess directional sensitivity rather than serve as fully optimized comparisons. We will add these additional results to the paper:
>
> 1. Context size: We co-fold with the full sequence, then crop to varying context sizes and re-fold on Runs N’ Poses. Performance with reduced context is stable across common crop sizes:
>
> | Context | Lig RMSD < 2Å (%) | RMSD < 2Å & LDDT-PLI > 0.8 (%) |
> |---|---|---|
> | Full sequence | 61.6 | 49.3 |
> | 256 tokens | 61.2 | 48.8 |
> | 196 tokens | 61.4 | 49.0 |
> | 164 tokens | 61.4 | 49.0 |
>
> 2. Distance bins: A model trained with a different bin configuration (1.2–50.75Å in log-space vs. 2–22Å linear) yields comparable ligand pose accuracy on the Foldbench benchmarks:
>
> | Bin setup | Lig RMSD < 2Å (%) | Prot RMSD < 2Å (%) | LDDT-PLI | LP-Entropy |
> |---|---|---|---|---|
> | 2–22Å linear (CoarseBind) | 52.0 | 65.8 | 0.599 | 0.470 |
> | 1.2–50.75Å log-space | 50.0 | 71.6 | 0.581 | 0.439 |
>
> 3. Validation splits: We follow the standard PDB time split with September 2021 date cutoff used across the field (Boltz-1, AlphaFold3, etc.). Additionally, FoldBench applies explicit sequence and structural similarity filtering, and our proprietary benchmarks are entirely unseen during training.
>
> > For continual learning, how does the pathwise GP approximation scale with observation count?
>
> The pathwise GP update requires a single matrix inversion scaling as O(N³) in the number of observations. In typical DMTA campaigns, observation counts remain low (<1,000 compounds), so this is not a practical bottleneck.
>
> > necessary and effiency of epinet, such as in many HTVS tailed distribution settings (activity cliff )?
>
> The epinet architecture is necessary because it provides joint predictive distributions across batches at negligible computational cost. This enables both the EMAX acquisition function and the continual learning scheme, neither of which is feasible with standard ensemble or dropout-based UQ without recomputing the expensive structural and affinity pairformer latents.
> Regarding long-tail distributions, the existing DMTA evaluation already tests the epinet on a pool of ~2,500 compounds for Target I, which exhibits harsh activity cliffs with a long-tailed potency distribution.
> We will add a binned potency histogram of the compound pool to make this explicit. The binned pIC50 counts for this target assay data are:
> | pIC50 Range | pIC50 < 6 | 6 < pIC50 < 7 | 7 < pIC50 < 8 | pIC50 > 8 |
> |---|---|---|---|---|
> | Count | 1601 | 432 | 587 | 68 |
>
>  Thus, the distribution roughly follows the usual long-tailed potency distributions found in realistic drug discovery campaigns.
> To quantify the activity cliffs present in this data, we can compare two activity cliff metrics at higher molecular similarity (Tanimoto > 0.7): (1) the IQR of pairwise |ΔpIC50|, and (2) the fraction of pairs exhibiting activity cliffs (|ΔpIC50| > 1). For Target I, we obtain IQR = 0.64 and activity cliff rate = 18%. We plan to show such pairwise pIC50 difference distributions in the paper for internal targets to better elucidate the difficulty of our internal data benchmarks (see response to reviewer AVsa).

---

> > ### Author Rebuttal · Reviewer_uUyB · 2026-04-05
> >
> > Thanks for the detailed and well-organized rebuttal, I will maintain my current score, as I believe the paper in its current form is already at a reasonable level and the promised revisions.

---

### Official Review · Reviewer_EpEj · 2026-03-13

**Soundness:** 3
**Presentation:** 3
**Significance:** 2
**Originality:** 1
**Overall Recommendation:** 4
**Confidence:** 2

**Summary:**

CoarseBind is a foundation model for protein-ligand binding affinity prediction that replaces expensive all-atom diffusion with a coarse structural representation(using only protein Cβ atoms and ligand heavy atoms )processed through a lean 48-layer Pairformer architecture. This design achieves 26× faster inference than Boltz-2 while outperforming it by 16–20% in Pearson correlation on both public (CASP16) and proprietary benchmarks, demonstrating that full atomic resolution is unnecessary for accurate affinity prediction. The model further incorporates an epistemic neural network (epinet) that provides well-calibrated uncertainty estimates, enabling risk-aware compound prioritization and a continual learning scheme that, through the EMAX acquisition function, achieves 6× greater affinity improvement over greedy selection strategies in simulated DMTA cycles. With only ~30M trainable parameters compared to ~509M in Boltz-2, CoarseBind offers a scalable and practical framework for high-throughput virtual screening and lead optimization campaigns.

**Compliance With Llm Reviewing Policy:**

Affirmed.

**Ethical Review Concerns:**

No comments.

**Final Justification:**

Most of my concerns have been addressed through the rebuttal.

**Key Questions For Authors:**

Why was Boltz-2 not included as a baseline in the experiments reported in Figures 3 and 4?

**Limitations:**

See weakness.

**Strengths And Weaknesses:**

**Summary of Concerns Regarding Novelty and Contribution**

The primary concern with this work lies in the limited architectural novelty. CoarseBind appears to be derived from existing co-folding architectures—most notably Boltz-2—by selectively removing the structure prediction components (i.e., the all-atom diffusion module) rather than introducing fundamentally new design principles. While the resulting efficiency gains are real and measurable, the path to achieving them raises questions about the degree of research contribution.

Specifically, substituting trainable encoders with frozen pretrained encoders is a well-established practice in the broader deep learning literature for reducing computational overhead, and does not itself constitute a meaningful research contribution in this context. Similarly, the Coarse Pose Module—as the authors themselves acknowledge—follows an implicit energy-based paradigm that is increasingly standard in the field, and its inclusion does not differentiate this work from contemporaneous approaches. The affinity architecture, beyond the likelihood estimation component, also appears largely consistent with prior work, with no clearly identified structural innovations.

Taken together, the overall impression is that CoarseBind represents an engineering effort to reduce computational cost through careful tuning of an existing, well-designed pipeline, rather than a research advance that opens new directions or introduces novel inductive biases.

That said, the Affinity Uncertainty Quantification component (Section 3.2.3) is a genuinely valuable contribution. Given that the unreliability of affinity predictions remains one of the central bottlenecks to the practical deployment of AI in industrial drug discovery, providing well-calibrated uncertainty estimates—particularly through the epinet-based likelihood module—addresses a pressing and underexplored gap. The demonstrated correlation between lower IQR and higher prediction success rates is meaningful, and the integration of this uncertainty module into a continual learning framework for DMTA cycles represents the most compelling and practically motivated aspect of this work.

---

> ### Author Rebuttal · Authors · 2026-03-30
>
> We thank the reviewer for the review. We respectfully disagree with the characterization of this work as primarily an engineering effort.
> > The primary concern with this work lies in the limited architectural novelty.
>
> We believe that focusing only on architectural additions is a too narrow view on scientific novelty.
> ICML's own criteria for originality states: "originality does not necessarily require introducing an entirely new method. Rather, a work that provides novel insights by evaluating existing methods, or demonstrates improved understanding is also equally valuable." It further asks whether the work "provides new insights, deepens understanding, or highlights important properties of existing methods" and whether it offers "a novel combination of existing techniques with well-articulated reasoning."
>
> We believe CoarseBind meets these criteria directly. The central contribution is the hypothesis (and its empirical validation) that full all-atom modeling and structure generation via diffusion is unnecessary for accurate binding affinity prediction. This is a non-obvious finding given that the dominant paradigm in the field (AlphaFold3, Boltz-1/2, NeuralPLexer3) treats iterative all-atom coordinate generation as essential. This constitutes a meaningful new insight, which has long-lasting implications for the field of molecular modeling, where coarse-grained models are still underused, especially for modeling binding.
>
> Beyond this central finding, the paper contributes several novel elements that the reviewer's summary does not fully account for:
> - Diffusion-free affinity prediction: Binding affinity prediction is one of the central computational bottlenecks in drug discovery. Current state-of-the-art models (Boltz-2) require ~28 seconds per complex, making library-scale screening of millions of compounds computationally intractable. CoarseBind reduces this to ~1 second per complex, directly unlocking high-throughput deployment that was previously infeasible. This is a qualitative shift in the scope that becomes practically possible in a virtual screening context.
> - Few-shot structural fine-tuning that improves affinity without retraining the affinity module: By fine-tuning only the structural Pairformer on as few as 3–6 proprietary crystals, CoarseBind achieves a ~17% affinity improvement on held-out compounds (with the affinity head frozen). This demonstrates a tight coupling between structural representation quality and affinity accuracy, and provides a practical mechanism for rapid program-specific adaptation that is absent from existing co-folding models.
> - Distogram entropy as a zero-shot affinity signal (H_LP), which correlates with both pose accuracy and binding affinity without any supervision, providing a model-intrinsic confidence metric that eliminates the need for a separate confidence module.
> - Structural evaluation with a diffusion-free pose module: The optimization-based coordinate generation scheme is applied here in a novel context: recovering coarse-grained poses from Pairformer distograms without any learned parameters. This enables structural evaluation and visualization while maintaining the speed advantage.
>
> > Why was Boltz-2 not included as a baseline in the experiments reported in Figures 3 and 4?
>
> Boltz-2 could not be included in the structure prediction benchmarks (Figures 3 and 4) because FoldBench, PoseBusters, and Runs N' Poses contain structures deposited to the PDB prior to Boltz-2's training cutoff (June 1, 2023), resulting in data leakage. We used Boltz-1 as the structural baseline, which shares the same training cutoff as CoarseBind (September 30, 2021). We note that Boltz-2 is included in the binding affinity benchmarks (Figure 5), which is the primary focus of this work.

---

> > ### Author Rebuttal · Reviewer_EpEj · 2026-04-05
> >
> > I agree with your point. I will raise the score.

---

### Official Review · Reviewer_AVsa · 2026-03-22

**Soundness:** 3
**Presentation:** 4
**Significance:** 3
**Originality:** 3
**Overall Recommendation:** 4
**Confidence:** 4

**Summary:**

This paper addresses the computational inefficiency of current deep learning-based protein-ligand binding affinity prediction methods (e.g., AlphaFold 3), which rely on expensive full-atom 3D coordinate generation. The authors argue that predicting binding affinity does not require full-atom structure generation — coarse-grained or local structural information suffices. They propose a lightweight architecture that extracts a local pocket representation from the Pairformer module to define the ligand-proximal binding pocket, and performs affinity prediction directly in latent space without explicit coordinate generation. An uncertainty-aware head is incorporated into the affinity module to provide calibrated confidence estimates. The resulting model achieves a significant speedup over full structure prediction pipelines while maintaining competitive accuracy.

**Compliance With Llm Reviewing Policy:**

Affirmed.

**Key Questions For Authors:**

See the weaknesses

**Limitations:**

yes

**Strengths And Weaknesses:**

Strengths

- The core observation that binding affinity prediction does not necessitate full-atom 3D coordinate generation is well-grounded. Decoupling affinity prediction from expensive structure generation is a practically meaningful direction, and the paper articulates this motivation clearly.

- Achieves 26x speedup compared to full structure prediction pipelines, which is a substantial practical improvement.

- The idea of extracting a ligand-proximal pocket representation from the Pairformer and performing prediction in latent space is clean and well-structured.

- The inclusion of an uncertainty-aware prediction head is a valuable addition.

- Clear presentation. The paper is generally well-written and the overall narrative is easy to follow.

Weaknesses

- Sounds like a reduction of an existing architecture. The proposed model is fundamentally built by removing components from AlphaFold's Pairformer that are designed for structure prediction, retaining only the parts relevant for affinity prediction. The resulting system is effective, the approach is more of an engineering subtraction than a methodologically novel contribution.

- The experimental evaluation would benefit from testing on broader and more challenging benchmark datasets, such as the Merck and JACS

---

> ### Author Rebuttal · Authors · 2026-03-30
>
> We thank the reviewer for the thoughtful review.
>
> > Sounds like a reduction of an existing architecture. The proposed model is fundamentally built by removing components from AlphaFold's Pairformer that are designed for structure prediction, retaining only the parts relevant for affinity prediction. The resulting system is effective, the approach is more of an engineering subtraction than a methodologically novel contribution.
>
> We believe the main scientific novelty of this paper is challenging the prevailing assumption in the field that binding affinity prediction requires all-atom resolution to accurately model protein-ligand interactions. Demonstrating that all-atom resolution is unnecessary has long-lasting implications for the field of molecular modeling (consider for instance the entire field of coarse-grained molecular modeling, which was so far deemed inappropriate for binding affinity prediction) and is therefore more than a simple re-engineering of an existing model.
>
> We also wish to clarify a factual point. The reviewer states that CoarseBind is "fundamentally built by removing components from AlphaFold's Pairformer that are designed for structure prediction." This is not entirely correct. The diffusion module in Boltz-2 samples atomistic structures, the most confident of which is used as a direct input to the Boltz-2 affinity module. The diffusion module is therefore a core part of the Boltz-2 affinity prediction pipeline, not merely an unrelated structure prediction component. The observation that this diffusion-generated structure is unnecessary for accurate affinity prediction is a non-obvious finding.
>
> > The experimental evaluation would benefit from testing on broader and more challenging benchmark datasets, such as the Merck and JACS
>
> Our evaluation already includes CASP16 (public, with 123 compounds in the main target) and 18 diverse proprietary targets (27,078 proprietary compounds, entirely unseen during training). We believe that these internal targets constitute a rigorous and challenging evaluation that better reflects real-world drug discovery conditions than public datasets (see e.g. recent two works that come to the same conclusion: [OpenFE private benchmark](https://chemrxiv.org/doi/full/10.26434/chemrxiv-2025-7sthd/v2), [Boltz-2 Recursion-internal assays](https://www.biorxiv.org/content/10.1101/2025.06.14.659707v1)). Hence, we did not focus on these public datasets when creating the affinity train-test split. Unfortunately, the Merck and JACS FEP benchmark assays overlap with our training data (via ChEMBL/BindingDB), precluding their use as held-out evaluation sets.
>
> Moreover, we note that the Merck and JACS benchmarks are curated congeneric series with fewer than 50 compounds per assay as they were originally designed for FEP validation, not for stress-testing ML models. To quantify this, we compare two activity cliff metrics at higher molecular similarity (Tanimoto > 0.7) for each set of target assay data: (1) the IQR of pairwise |ΔpIC50|, and (2) the fraction of pairs exhibiting activity cliffs (|ΔpIC50| > 1). For the combined Merck/JACS sets (averaging over target assays with more than 20 compounds each): IQR = 0.63, activity cliff rate = 18%. For CASP16: IQR=0.51, activity cliff rate = 18%. For our proprietary targets: IQR = 0.82, activity cliff rate = 25%. This confirms that our proprietary evaluation contains more and harsher activity cliffs and is therefore already more challenging than the suggested FEP benchmarks. We will add figures regarding this comparison of public and private potency distributions to the article.

---

> > ### Author Rebuttal · Reviewer_AVsa · 2026-04-08
> >
> > Thank you to the authors for the response. I believe that the dataset construction process and its details should be further elaborated in the main text or the appendix.

---

### Decision · Program_Chairs · 2026-04-30

**Decision:**

Accept (regular)

**Comment:**

The paper proposes CoarseBind, a protein–ligand binding framework that replaces full all-atom diffusion with a coarse pocket-level representation and a diffusion-free pipeline for pose and affinity prediction, with the goal of improving screening throughput while retaining useful structural signal. The main empirical claim is that this lighter design remains competitive on pose benchmarks and achieves stronger affinity prediction than Boltz-2 on the reported CASP16 and proprietary assay evaluations, while also providing uncertainty estimates that may help prioritization. Overall, the work is technically clear and the efficiency angle is practically relevant.